# Transcriptional profiling of sequentially generated septal neuron fates

**Miguel Turrero García[1]\*[†], Sarah K Stegmann[1], Tiara E Lacey[1,2],
Christopher M Reid[1,3][†], Sinisa Hrvatin[1], Caleb Weinreb[4,5], Manal A Adam[1][†],
M Aurel Nagy[1,3], Corey C Harwell[1]\*[†]**

[1]Department of Neurobiology, Harvard Medical School, Boston, United States;
[2]Biological and Biomedical Sciences PhD program at Harvard University, Cambridge,
United States; [3]PhD Program in Neuroscience at Harvard University, Cambridge,
United States; [4]Department of Systems Biology, Harvard Medical School, Boston,
United States; [5]PhD Program in Systems Biology at Harvard University, Cambridge,
United States

**\*For correspondence:**
Miguel.TurreroGarcia@ucsf.edu
(MTG);
Corey.Harwell@ucsf.edu (CCH)

**Present address:** [†]Department
of Neurology, University of
California San Francisco, San
Francisco, United States

**Competing interest:** The authors
declare that no competing
interests exist.

**Reviewing Editor:** Joseph
G Gleeson, Howard Hughes
Medical Institute, The Rockefeller
University, United States

**Abstract** The septum is a ventral forebrain structure known to regulate innate behaviors. During
embryonic development, septal neurons are produced in multiple proliferative areas from neural
progenitors following transcriptional programs that are still largely unknown. Here, we use a combi-
nation of single-cell RNA sequencing, histology, and genetic models to address how septal neuron
diversity is established during neurogenesis. We find that the transcriptional profiles of septal
progenitors change along neurogenesis, coinciding with the generation of distinct neuron types. We
characterize the septal eminence, an anatomically distinct and transient proliferative zone composed
of progenitors with distinctive molecular profiles, proliferative capacity, and fate potential compared
to the rostral septal progenitor zone. We show that *Nkx2.1*-expressing septal eminence progenitors
give rise to neurons belonging to at least three morphological classes, born in temporal cohorts that
are distributed across different septal nuclei in a sequential fountain-like pattern. Our study provides
insight into the molecular programs that control the sequential production of different neuronal
types in the septum, a structure with important roles in regulating mood and motivation.

## Editor's evaluation

This paper captures the developmental trajectories of septal neurons and unravels their genetic
codes. The authors use both single cell sequencing and transgenic mice to quantify septal cells
derived from two septal progenitor zone, assessing the temporal dynamics of septal cells generated
from one of these zones and classify the neurons according to morphology. The work will be of
interest to developmental neurobiologists studying forebrain neurogenesis. The work is novel and
provides substantial molecular insight into transcriptomic profiles of cells in the septum. The work
will serve as a reference for future brain networks analysis.

## Introduction

A central question in developmental neurobiology is how a pool of seemingly uniform embryonic
neural stem cells can generate the immense diversity of neuronal subtypes in the mature brain.
During mammalian brain development, neural progenitors acquire distinctive spatial identities and
progress through a series of temporal competence states to give rise to their neuronal and glial
progeny (*Holguera and Desplan, 2018*; *Ohtsuka and Kageyama, 2019*). There has been significant
progress in defining the key molecular differences between spatially segregated progenitor domains

(*Azzarelli et al., 2015*). For example, the transcription factors EMX1 and DLX2 define the pallial and subpallial progenitor domains responsible for the production of glutamatergic and GABAergic forebrain neurons, respectively (*Puelles et al., 2000*). Temporal programming of forebrain progenitors is perhaps best exemplified in the developing cerebral cortex, where neurons are sequentially produced to populate the six cortical layers in an inside-out pattern (*Lodato and Arlotta, 2015*). One of the major challenges in the field has been to identify the molecular programs controlling the timely generation of diverse neuronal subtypes from each progenitor region within the forebrain (*Kohwi and Doe, 2013*). Recent advances in single-cell RNA sequencing (scRNA-Seq) techniques have greatly increased our understanding of the molecular identity of numerous mature neuronal types (*Yuste et al., 2020*) and began to address the dynamic transcriptional changes underlying their specification (*Loo et al., 2019*). In the ventral forebrain, several studies have used scRNA-Seq to address how and when GABAergic neuronal diversity arises (*Mayer et al., 2018*; *Mi et al., 2018*) and to obtain an increasingly clear picture of how hypothalamic development is orchestrated (*Kim et al., 2020*; *Romanov et al., 2020*; *Zhou et al., 2020*). In spite of this progress, the molecular programs regulating the production of basal forebrain projection neurons in other proliferative zones remain poorly understood.

The septum is a ventral forebrain structure composed of a diverse array of GABAergic, cholinergic, and glutamatergic projection neurons that regulate a range of innate behaviors governing emotion and affect (*Sheehan et al., 2004*). The mature septum is segregated into a complex formed by the medial septum and the diagonal band of Broca (MS/DBB, hereafter referred to as 'MS' for simplicity) and lateral septum (LS) nuclei (in turn subdivided into dorsal, intermediate, and ventral nuclei [LSd, LSi, and LSv, respectively]), which have distinctive efferent and afferent connectivity with a variety of other areas of the brain (*Sheehan et al., 2004*). Most septal neurons are generated from neural progenitors located in two portions of the embryonic brain: the septum proper, which develops between the lateral ganglionic eminence (LGE) and the most anterior part of the cortex, and a small and transient proliferative region adjacent to the medial ganglionic eminence (MGE) and rostral with respect to the embryonic preoptic area (PoA). The expression of Zic family transcription factors distinguishes septal progenitor zones from adjacent regions (*Inoue et al., 2007*). The developing septum can be further divided into pallial-like, LGE-like, and MGE/PoA-like areas demarcated by the enriched expression of transcription factors such as *Tbr1*, *Gsh2*, and *Nkx2.1*, respectively, which give rise to specific subpopulations of neurons (*Flames et al., 2007*; *Iyer and Tole, 2020*; *Puelles et al., 2000*; *Wei et al., 2012*). The anatomically distinct MGE/PoA-like region has been previously described as the 'ventral septum' (vSe) to differentiate it from the septum proper (*Flames et al., 2007*; *Hoch et al., 2015b*; *Hoch et al., 2009*; *Puelles et al., 2000*; *Wei et al., 2012*). Given its transient nature and similarities to the ganglionic eminences, we refer to this region throughout this article as the septal eminence. Temporal production of neurons in the septum follows a general medial-to-lateral pattern: MS neurons are born earlier in development, while later-born cells occupy progressively more lateral positions in the LS (*Creps, 1974*; *Wei et al., 2012*). Despite some recent progress in understanding the spatial and temporal origins of diverse neuronal subgroups in the septum (*Iyer and Tole, 2020*; *Magno et al., 2017*; *Wei et al., 2012*), little is known about the molecular programs guiding temporal competence states of septal progenitors and how they lead to the specification of different types of neurons. In this study, we address this by performing scRNA-Seq on septa at different stages in development to infer the developmental trajectories connecting progenitors to the neurons derived from them. We generate a comprehensive dataset and interrogate it to gain insight into the extent neuronal diversity in the mature septum and how it is generated during development. We focus on two stages at the peak of MS and LS neurogenesis and find that the transcriptomic profile of neural progenitors and newborn neurons changes greatly between early and late neurogenic periods. Through genetic fate-mapping experiments, we resolve the contribution of progenitors located in the septum proper and the septal eminence to mature neuronal diversity. Finally, we describe the temporal pattern for the generation of diverse neuron types derived from the septal eminence defined by their morphological features and allocation within the septum. Distinctive subsets of septal neurons are important for regulating specific aspects of emotional and affective behavioral states. This study provides a comprehensive molecular framework for identifying the candidate gene networks involved in determining spatially and temporally defined septal neuron fates.

## Results

### Early emergence of distinct neuronal lineages in the developing septum

We selected six ages spanning the entire development of the septum, from the embryonic period to its maturity. We chose developmental stages that would cover the peak period of neurogenesis for both the medial (embryonic day [E]11) and lateral septum (E14) (*Wei et al., 2012*), as well as the subsequent processes of neuronal maturation, such as migration, axonal outgrowth, and synaptogenesis (E17, postnatal days [P]3 and 10), along with P30 as a stage representative of a mature septum (*Figure 1A*, top). For each developmental stage, we manually dissected the septum (for E11, E14, and E17, the MGEs/PoAs, as potential sources of septal neurons [*Wei et al., 2012*], were collected as well), generated a single-cell suspension, and subjected it to single-cell encapsulation followed by RNA sequencing using a custom inDrops microfluidic system (*Klein et al., 2015*; *Zilionis et al., 2017*; *Figure 1A*, bottom). After filtering the data based on quality control criteria (see Materials and methods), we obtained a total of 72,243 cells across the six developmental stages. Plotting the data using UMAP, a method that yields two-dimensional graphic representations where the position of individual cells conveys information about their relationships, we identified clusters corresponding to radial glia (RG), intermediate progenitors (IP), and neurons (N) based upon marker expression (*Figure 1B*). These identities were largely consistent with t-distributed stochastic neighbor embedding (t-SNE) of the same data (*Figure 1—figure supplement 1A*), where cluster identities were assigned based on an extensive set of putative marker genes (*Figure 1—figure supplement 1B*). As expected from the experimental design, our dataset contains a number of distinct cell types, including different progenitor classes (radial glia and intermediate progenitors) and neurons at progressive stages of maturation (newborn, migrating, wiring, and mature), as well as a number of other cells that were not the focus of this study (glia, ependymal cells, endothelial cells, etc.) (*Figure 1—figure supplement 1A and B*). However, in order to resolve potential relationships among individual cells and thus understand the molecular changes happening along defined lineages, we needed a tool that allowed better visualization of inferred trajectories. We decided to use SPRING, a tool that generates a k-nearest-neighbor layout where each cell is represented as a node extending edges to the 'k' other nodes within the dataset with most similar gene expression profiles (*Weinreb et al., 2018*). This resulted in a graph containing all cells we sequenced, which were aligned according to their developmental stage despite the fact that SPRING was agnostic to the origin of each cell (*Figure 1C*). The clusters that we had identified could be projected onto the SPRING visualization, confirming the similarities between cells located close to each other within the plot (*Figure 1—figure supplement 2A*). Based on gene expression, we could distinguish groups of cells corresponding to the glial lineage, organized along developmental trajectories from progenitors to mature astrocytes (*Figure 1—figure supplement 2B*) and oligondendrocytes (*Figure 1—figure supplement 2C*). To keep the focus on septal neuron specification, we removed cells identified as non-neuronal (*Figure 1—figure supplement 1A and B*) and produced a SPRING plot containing 53,011 cells that includes neurons at different maturation stages as well as the progenitors that gave rise to them (*Figure 1D*). We then produced SPRING plots of the E11 and E14 timepoints, corresponding respectively to the peak medial and lateral septal neurogenic periods. (*Figure 1E and F*). We used the genes *Nes* (nestin), *Ascl1* (achaete-scute family bHLH transcription factor 1), and *Dcx* (doublecortin) as cell-type markers for radial glia, intermediate progenitors, and newborn neurons, respectively (*Gleeson et al., 1999*; *Lendahl et al., 1990*; *Yun et al., 2002*; *Figure 1E and F*). The SPRING visualizations for E11 and E14 had a consistent organization, where more stem-like cells (i.e., radial glia) were located at one end of the plot, and more differentiated ones (i.e., neurons) at the opposite, with intermediate progenitors interspersed between them, reminiscent of the known lineage relationships among these three cell types (*Figure 1B and C*). Sectors of the SPRING plots comprised newborn neurons were sharply divided into distinct protruding clusters, representing cells with common molecular identities. We assigned prospective identities to each of those clusters based upon their marker gene expression and confirmed the presence of cells expressing the said markers during MS (E11) and LS (E13, E14) neurogenic stages using the Allen Developing Mouse Brain Atlas (*Figure 1G and H*, *Figure 1—figure supplement 2D and E*). MGE/PoA-derived neurons were identified based on their expression of *Lhx6* (LIM homeobox 6) (*Alifragis et al., 2004*), while newborn cholinergic neurons expressed *Gbx2* (gastrulation brain homeobox 2) (*Asbreuk et al., 2002*; *Chen et al., 2010*); we also identified *Dner* (delta/notch-like EGF repeat containing) as a general

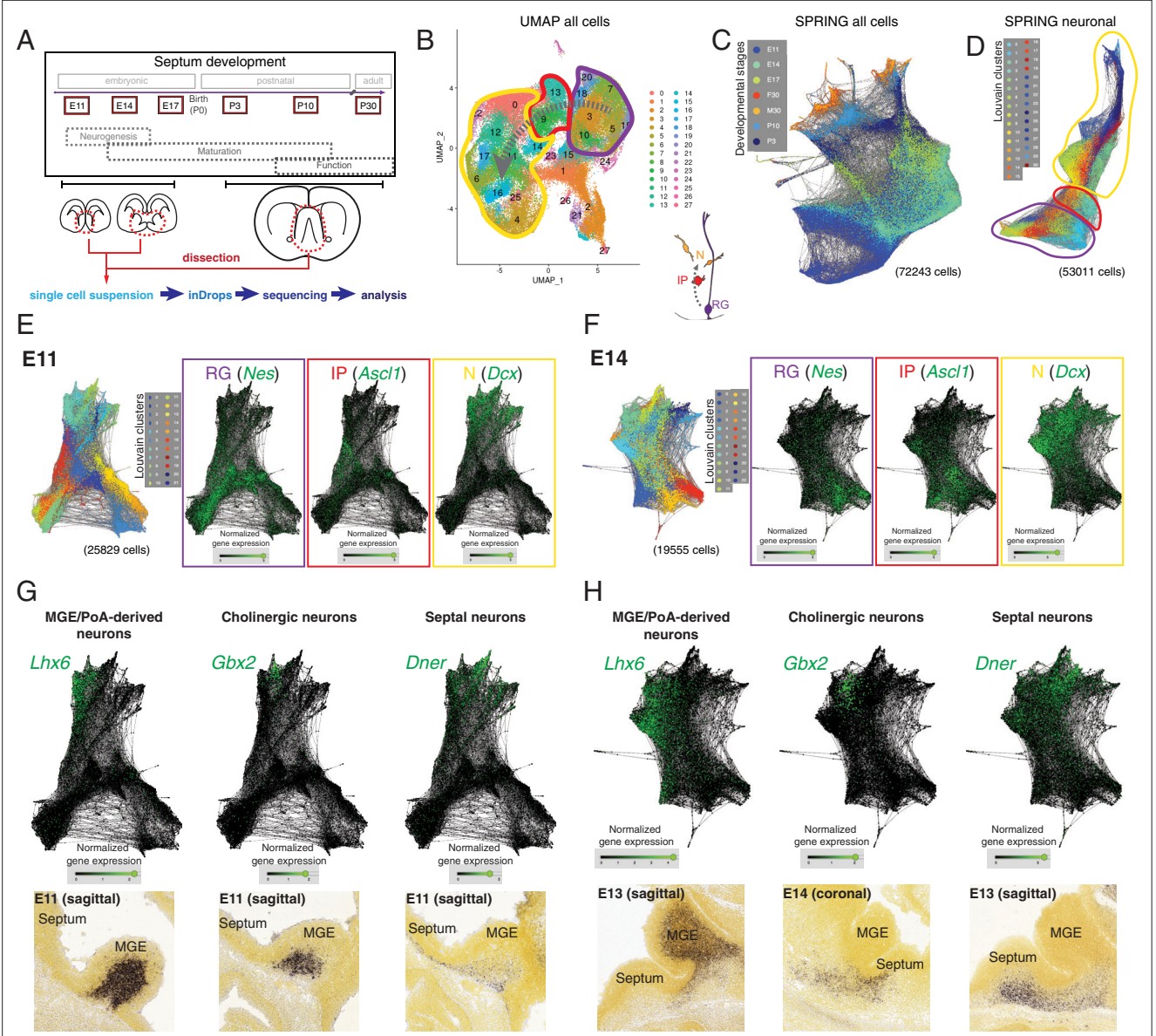

**Figure 1.** Single-cell RNA sequencing of the developing septum reveals early emergence of neuronal identities. (**A**) Experimental approach: samples were collected at the indicated developmental stages and submitted to single-cell RNA sequencing using inDrops. (**B**) UMAP plot of all cells, where the transition from radial glia (RG, purple) to neurons (N, yellow) via intermediate progenitors (IP, red) schematized in the cartoon at the bottom right can be visualized (gray dashed arrows). (**C**) SPRING plot shows developmental stage-dependent organization of all sequenced cells. (**D**) SPRING plot of cells belonging to neuronal trajectory (colors indicate cell groups determined by Louvain clustering). (**E, F**) SPRING plots of all cells at embryonic stages E11 (**E**) and E14 (**F**) display differentiation-dependent alignment of cells, illustrated by the relative enrichment of the genes *Nes*, *Ascl1*, and *Dcx* (cell-type markers for RG, IP, and N, respectively) in adjacent areas of the graphs. (**G, H**) Analysis of each protrusion within the neuronal portion of E11 (**G**) and E14 (**H**) SPRING plots shows enrichment in marker genes for MGE/PoA-derived (*Lhx6*), cholinergic (*Gbx2*), and septal (*Dner*) newborn neurons; the presence of these neuronal types in the embryonic septum at similar stages is confirmed by in situ hybridization (bottom panels).

Image credit (bottom panels): Allen Institute – Allen Developing Mouse Brain Atlas.

The online version of this article includes the following figure supplement(s) for figure 1:

**Figure supplement 1.** Cluster identity assignment in the scRNA-Seq dataset.

**Figure supplement 2.** Additional cell identities in the SPRING plot.

**Figure supplement 3.** Analysis of postnatal day (P)30 dataset for discovery of novel septal neuron markers.

marker for septal neurons (*Figure 1G and H*). We identified additional distinct newborn neuronal populations, including presumptive pallium-derived neurons expressing *Tbr1* (T-box brain transcription factor 1) (*Bulfone et al., 1995*), septum-derived Cajal-Retzius cells (*Bielle et al., 2005*) expressing *Trp73* (transformation-related protein 73) (*Causeret et al., 2021*; *Meyer et al., 2004*), and presumptive LGE-derived neurons expressing *Isl1* (ISL LIM homeobox 1) (*Stenman et al., 2003*; *Toresson et al., 2000*; *Figure 1—figure supplement 2D and E*). The expression of these genes was restricted to the mantle zones of the MGE and the developing septum, further confirming that they are markers of postmitotic neurons, rather than progenitors (*Figure 1G and H*, *Figure 1—figure supplement 1C*, *Figure 1—figure supplement 2*). While we recovered a relatively low number of cells from P30 samples (*Figure 1—figure supplement 3A*), and only a fraction of these (23%) could be identified as MS or LS neurons, analysis of those groups of cells allowed us to identify several previously unreported potential markers for neurons located in either the LS or MS (*Figure 1—figure supplement 3B*), or common to both nuclei (*Figure 1—figure supplement 3C*). We identified *Prkcd* (protein kinase C delta) as a potential LS marker gene and confirmed its restricted expression by crossing a Prkcd-Cre mouse line (*Kalish et al., 2018*) with a Cre-dependent reporter line (*Figure 1—figure supplement 3D*). Nearly all labeled cells were neurons confined to the LS (*Figure 1—figure supplement 3E*), and thus could be assumed to be largely GABAergic (*Zhao et al., 2013*). As far as we are aware, this is the first report of a mouse line that grants wide genetic access to GABAergic neurons in the LS. Together, these findings demonstrate that our scRNA-Seq dataset can be used to identify diverse molecular cell types and infer their developmental trajectories within the developing septum.

## The septal eminence as a specialized proliferative area

Previous work has shown that a portion of cells in the septum, most notably MS cholinergic neurons, are derived from progenitors expressing the transcription factor *Nkx2.1* (*Magno et al., 2017*; *Wei et al., 2012*; *Xu et al., 2008*). Within the forebrain, *Nkx2.1* is expressed in the MGE (*Sussel et al., 1999*) as well as in the PoA and in the caudal portion of the developing septum (*Magno et al., 2017*; *Puelles et al., 2000*; *Rubin et al., 2010*). We hypothesized that the caudal portion of the embryonic septum (i.e., the region we define as the septal eminence) could be the developmental source of several types of mature septal neurons. Since our embryonic samples contained cells from both the MGE/PoA and the septum, as identified in SPRING plots (*Figure 1G and H*), we needed an additional marker to distinguish between these proliferative areas. We analyzed the expression of *Zic4*, a general septal marker (*Rubin et al., 2010*), and *Nkx2.1* in the E14 SPRING plot, revealing a region of overlapping expression spanning progenitors and newborn neurons (*Figure 2A*). We confirmed that embryonic progenitors located in the VZ/SVZ proliferative areas of the caudal, but not the rostral, septum were positive for NKX2.1 (*Magno et al., 2017*; *Figure 2B*). Using embryonic samples of Nkx2.1-Cre;Ai14 mice, a model where cells with a developmental history of *Nkx2.1* expression are labeled by the fluorescent reporter tdTomato, in combination with immunofluorescence staining with an antibody that recognizes several ZIC isoforms, including ZIC4 (*Borghesani et al., 2002*), we found a stream of ZIC-positive cells migrating rostrally from their putative site of origin in the caudal septum, consistent with previously described tangential migration of *Nkx2.1*-lineage cells into the septum from caudal locations (*Wei et al., 2012*; *Figure 2C*). The distinct neuronal output from the septal eminence and its expression of *Nkx2.1* suggests that there may be fundamental differences in progenitor composition and proliferative behaviors between the rostral and caudal proliferative zones in the embryonic septum (*Magno et al., 2017*). To address these potential differences, we performed immunostaining for the mitotic marker phosphorylated histone 3 on E13 Nkx2.1-Cre;Ai14 samples in order to compare the abundance and location of cycling progenitors between the rostral and caudal (*Nkx2.1*-expressing) portions of the developing septum (*Figure 2D*). This experiment revealed a much lower proportion of dividing cells located in the putative subventricular zone of the septal eminence when compared to the rostral portion of the septum (*Figure 2D and E*). This could reflect underlying cell biological differences between these two regions, whereby fate-restricted progenitors in the septal eminence would preferentially undergo direct neurogenic divisions at the ventricular surface rather than delaminating and entering the subventricular zone as transit-amplifying intermediate progenitors (*Petros et al., 2015*; *Turrero García and Harwell, 2017*). We performed immunofluorescence staining for NKX2.1 together with ASCL1 as a marker for fate-committed, potentially terminally dividing progenitors (*Yun et al., 2002*) on E14 samples; we found a roughly twofold increase in the

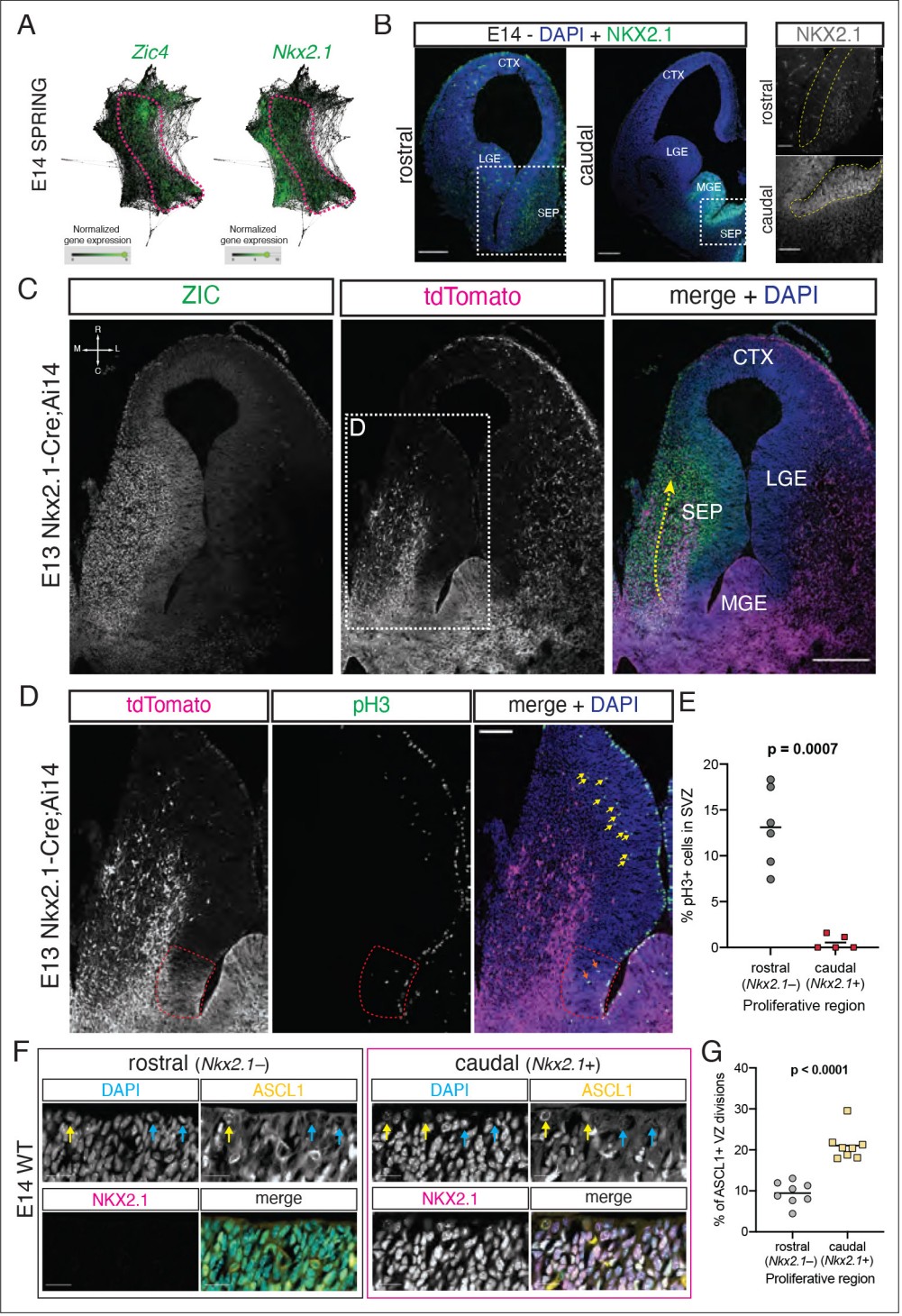

**Figure 2.** Neural progenitors within the septal eminence preferentially divide at the ventricular surface. (**A**) SPRING plots at embryonic day (E)14 show that a set of cells at this stage (dashed magenta line) express both *Zic4* (left) and *Nkx2.1* (right). (**B**) Immunofluorescence staining for NKX2.1 (green; counterstained with DAPI, blue) at rostral and caudal locations of the septum on coronal sections of an E14 brain. Panels on the right display magnified insets (marked with dashed white squares on the corresponding overview images); the main proliferative area (ventricular zone) is highlighted by a yellow dashed line, showing NKX2.1-positive cells in the caudal, but not rostral, septal anlage. Scale bars: 250 µm (overview images); 100 µm (close-up images). (**C**) Horizontal section (compass on the top left of left panel indicates rostrocaudal and mediolateral axes) of the right hemisphere of an E13 Nkx2.1-Cre;Ai14 mouse brain. Immunofluorescence staining for ZIC proteins (left panel; green in merge) and the tdTomato

*Figure 2 continued on next page*

*Figure 2 continued*

fluorescent reporter (middle panel; magenta in merge), shown both as single channels and merged (right panel, with DAPI counterstaining in blue), show a subset of *Nkx2.1*-expressing septal progenitors in the septal eminence, as well as a caudal-to-rostral stream of migrating ZIC-positive neurons with a developmental history of *Nkx2.1* expression (yellow dashed arrow). Scale bar, 250 µm. (**D**) Close-up of the area indicated by the dashed white line in (**C**); immunofluorescence staining for tdTomato (magenta in merge) and phosphorylated histone 3 (pH3, green in merge) shows difference in the number of subapically dividing cells in the rostral portion of the septum (yellow arrows) compared to the *Nkx2.1*-expressing (red dashed line) septal eminence (orange arrows). Scale bar, 100 µm. (**E**) Quantification of the proportion of dividing (pH3+) cells located in the subventricular zone of the rostral (gray dots) and caudal (red squares), that is, *Nkx2.1*– and *Nkx2.1*+, proliferative regions of the developing septum at E13. (**F**) Immunostaining for ASCL1 (yellow in merge) and NKX2.1 (magenta in merge), counterstained with DAPI (blue in merge), in rostral and caudal proliferative regions of the septum of an E14 mouse brain, highlighting ASCL1+ (yellow arrows on DAPI and ASCL1 panels) and ASCL1– (blue arrows on DAPI and ASCL1 panels) dividing cells at the apical surface. Scale bars, 20 µm. (**G**) Quantification of the proportion of ventricular surface divisions that are ASCL1+ in the rostral (gray dots) and caudal (yellow squares) proliferative regions of the developing septum at E14. All data points are represented; black bars represent the mean. Unpaired t-tests were performed; the p-values are indicated above the corresponding compared sets of data: bold typeface indicates statistically significant differences (p<0.05). CTX: cortex; LGE: lateral ganglionic eminence; MGE: medial ganglionic eminence; SEP: septum.

The online version of this article includes the following source data for figure 2:

**Source data 1.** Quantifications of septal eminence progenitors.

proportion of ASCL1+ mitotic cells at the ventricular surface of the NKX2.1+ portion of the septal anlage (*Figure 2F and G*). Together, these data suggest that the caudal portion of the developing septum is composed of progenitors with distinctive molecular profiles, proliferative capacity, and fate potential compared to its rostral counterpart. Given the similarities between the caudal developing septum and the ganglionic eminences and its distinctive progenitor composition, we propose to name this proliferative zone as the septal eminence (see 'Discussion').

## Fate mapping of neurons derived from the septal eminence

To better understand the contribution of rostral and septal eminence progenitors to the mature complement of septal neurons, we used three different genetic mouse models to fate-map P30 septal cells derived from *Zic4*-expressing (all septal proliferative areas), *Nkx2.1*-expressing (MGE/PoA and septal eminence), or *Zic4* and *Nkx2.1* coexpressing (septal eminence) progenitors (*Figure 3A, C and E*). Given the differences in septal anatomy along the rostrocaudal axis (*Creps, 1974*; *Figure 3— figure supplement 1A*), we quantified cells at rostral (R), medial (M), and caudal (C) locations. Combining a Zic4-Cre driver line (*Rubin et al., 2010*) with the nuclear membrane fluorescent reporter Sun1-GFP (*Mo et al., 2015*; *Figure 3A*), we found that the vast majority of neurons (71–97%) in the three main subdivisions of the LS (dorsal, intermediate, and ventral nuclei), and approximately 50% of MS neurons, had a developmental history of *Zic4* expression throughout the rostrocaudal axis (*Figure 3B*). We then used the Nkx2.1-Cre;Ai14 mouse model (*Figure 3C*), in combination with ZIC immunofluorescence staining, to understand the relative contribution of *Nkx2.1*-expressing progenitors to the mature septum-derived neuronal population. About 10–30% of ZIC+ cells in both the LS and the MS, in decreasing proportion along the rostrocaudal axis, had a developmental history of *Nkx2.1* expression (*Figure 3D*). To further refine our analyses, we used an intersectional genetic approach, combining an Nkx2.1-IRES-FlpO (Nkx2.1-Flp) knock-in driver line (*He et al., 2016*) with a Zic4-Cre driver line and the FLTG reporter line (*Plummer et al., 2015*; *Figure 3E*). Cells with a history of expression of both genes (intersectional population, septal eminence-derived) would result in labeling with GFP, while expression of only *Nkx2.1* (subtractive population, MGE/PoA-derived) would label cells with tdTomato (*Plummer et al., 2015*). The intersectional population was predominantly located in the LS, allocated to progressively more ventral locations along the rostrocaudal axis, while the subtractive population was mainly found in the MS (*Figure 3F*, left). We performed immunofluorescence staining for the neuronal marker NeuN (*Figure 3—figure supplement 1B and C*) and found that the vast majority of the intersectional population, but only a small fraction of the subtractive cells, were neurons (*Figure 3F*, right). This observation was supported by the distinct astrocytic morphology of many MS subtractive cells (*Figure 3E*, *Figure 3—figure supplement 1A–C*), and it

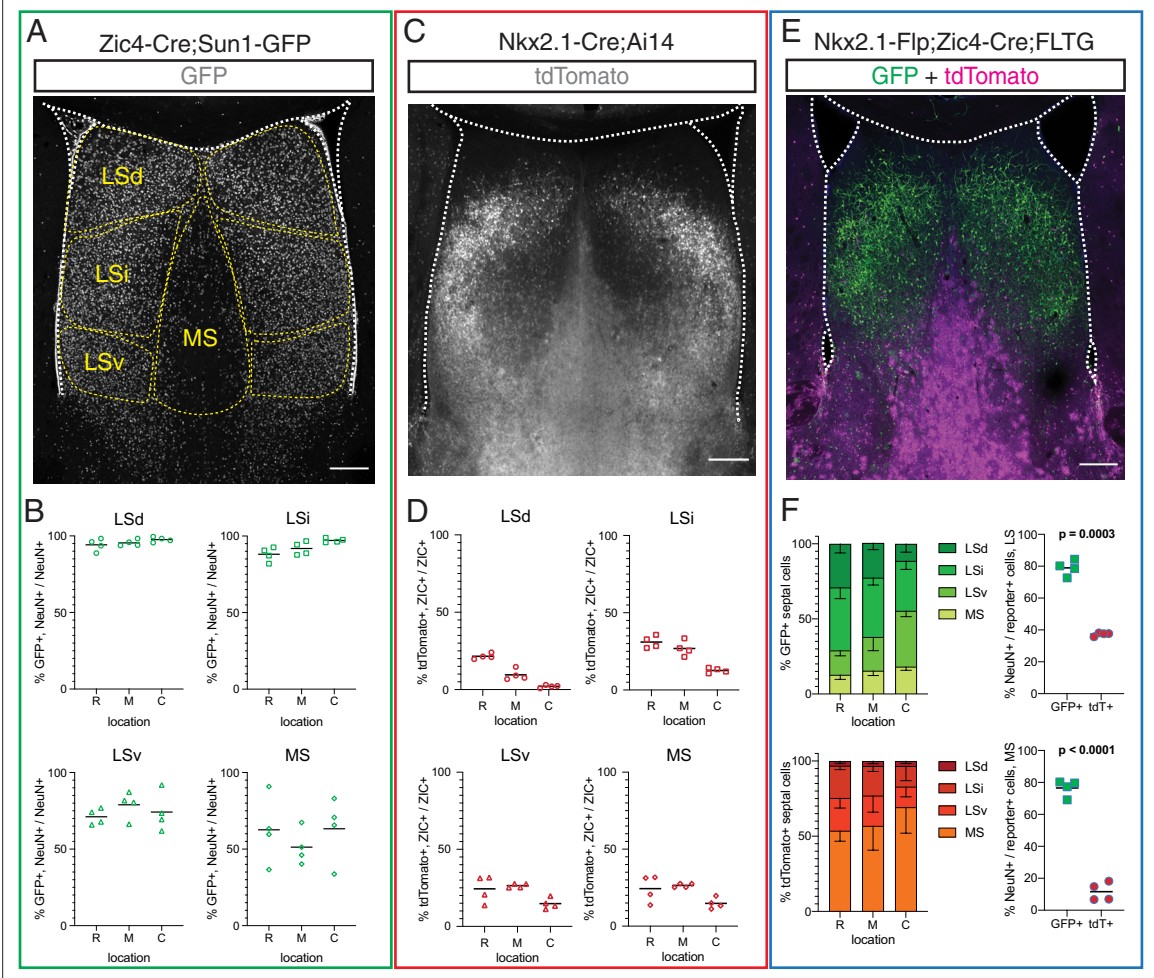

**Figure 3.** Fate mapping of septal eminence-derived neurons in the mature septum. (**A**) Coronal section of the septum of a postnatal day (P)30 Zic4-Cre;Sun1-GFP mouse; immunofluorescence staining for the reporter GFP shows the location of *Zic4*-lineage cells. Yellow dashed lines indicate the location of medial septum (MS) and lateral septum (LS) nuclei as indicated. (**B**) Quantification of the proportion of neurons within the *Zic4*-lineage within the total NeuN+ neuronal population. (**C**) Coronal section of the septum of a P30 Nkx2.1-Cre;Ai14 mouse; immunofluorescence staining for the reporter tdTomato shows the location and morphology of *Nkx2.1*-lineage cells. (**D**) Quantification of the proportion of *Nkx2.1*-lineage cells within the ZIC+ population, as recognized by a pan-ZIC antibody. (**E**) Coronal section of the septum of a P30 Nkx2.1-Flp;Zic4-Cre;FLTG mouse; immunofluorescence staining for the reporters GFP (green) and tdTomato (magenta) shows the location and morphology cells within the *Nkx2.1* lineage with (GFP+) or without (tdTomato+) a developmental history of *Zic4* expression. (**F**) Left: proportion of the entire intersectional (GFP+, top) or subtractive (tdTomato+, bottom) populations allocated within each septal nucleus in the Nkx2.1-Flp;Zic4-Cre;FLTG line, as illustrated in (**C**); right: proportion of cells positive for the neuronal marker NeuN within the intersectional (GFP+, green squares) and subtractive (tdTomato+, red circles) populations in the entire LS (top) and in the MS (bottom). Scale bars (**A, C, E**), 250 μm. Quantifications in (**B**), (**D**), and (**F**) were obtained for each of the mouse lines above the corresponding set of graphs, across the dorsal, intermediate, and ventral nuclei of the LS (LSd, LSi, and LSv, respectively) and the MS, at rostral (R), medial (M), and caudal (C) locations along the rostrocaudal axis. Unpaired t-tests (right graphs in **F**) were performed; the p-values are indicated above the corresponding compared sets of data: bold typeface indicates statistically significant differences (p<0.05).

The online version of this article includes the following source data and figure supplement(s) for figure 3:

**Source data 1.** Fate mapping of septal eminence derived cells.

**Figure supplement 1.** Neuronal identity within the intersectional and subtractive populations in the septum Nkx2.1-Flp;Zic4-Cre;FLTG mice.

**Figure supplement 1—source data 1.** Percentage of neurons within fate-mapped populations.

was true for both the LS and the MS, with only minor changes along the rostrocaudal axis (*Figure 3—figure supplement 1D and E*). Our results demonstrate that the septal eminence generates neurons that largely occupy the LS, while cells derived from the MGE/PoA are mainly allocated to the medial septum and appear to be glia rather than neurons, consistent with previous reports (*Wei et al., 2012*).

## Temporal transcriptional programs during MS and LS neurogenesis

After confirming the existence of at least two spatially distinct developmental origins for septal neurons, we addressed the crucial temporal component of neuronal fate determination (*Kohwi and Doe, 2013*). Neurons destined for the medial septum are mainly generated during early neurogenesis, while those that will occupy the LS are born at later stages (*Wei et al., 2012*). We reasoned that since neuronal subtypes can already be distinguished at embryonic stages (*Figure 1G and H*, *Figure 1—figure supplement 2D and E*), different molecular programs guiding the generation of MS versus LS neurons from neural progenitor cells would be detectable as well. We used our scRNA-Seq dataset to compare the gene expression profiles of each of the three main cell types present along the neurogenic sequence (radial glia, intermediate progenitors, and newborn neurons) across the three embryonic stages we had collected and found numerous genes that were differentially expressed (*Figure 4A*). Several classes of genes showed different levels of expression across developmental stages, including transcription factors, cell adhesion molecules, and intercellular and intracellular signaling molecules (*Figure 4—figure supplement 1A–D*). We focused on a select subset of genes composed of cell-type markers and candidate temporal competence factors (*Figure 4B*). Since neurogenesis is largely completed by E17 and signatures of gliogenesis could already be detected, we decided to focus our subsequent analyses on the neurogenic stages E11 and E14. To validate our in silico differential gene expression analysis, we performed single-molecule in situ hybridization (RNAscope) for selected mRNA transcripts and quantified the signals during early (E12) and late (E14) septal neurogenesis (*Figure 4C–H*). We selected genes that were differentially expressed either at E11 or E14 and compared their expression in the septum proper (rostral) and the septal eminence (caudal) within the relevant zone demarcated by the expression of cell-type marker genes (*Nes* for RG, *Ascl1* for IP, and *Dcx* for NN). Transcripts for genes of interest were normalized to the relevant marker gene. In RG, *Hmga2* was significantly enriched in E12 septum (*Figure 4C and F*), while *Hes5* was enriched at E14 (*Figure 4—figure supplement 1E*). In IP, both *Ccnd2* (*Figure 4D and G*) and *Ccnd1* (*Figure 4—figure supplement 1F*) were significantly enriched at E14. The gene *Prdm16*, a known marker of RG in other areas of the telencephalon (*Baizabal et al., 2018*; *Shimada et al., 2017*), was clearly upregulated in late-born neurons (*Figure 4E and H*); the same was true for other markers generally expressed at later neurogenic stages, such as *Nfia* (*Clark et al., 2019*; *Figure 4—figure supplement 1G*). Together, our data provide a framework for characterizing dynamic gene expression as progenitors transition from generating MS neurons to LS neurons.

## Morphology and distribution of temporal cohorts of neurons in the *Nkx2.1* lineage

Septal neurons are born in a defined medial-to-lateral temporal sequence, where the MS is generated early in neurogenesis and later-born neurons are allocated to progressively more lateral positions (*Creps, 1974*; *Wei et al., 2012*). Our molecular profiling data suggest that the temporally defined molecular states of septal progenitors and neuronal precursors correlate with the production of specific cell fates. To better understand how birthdate affects the specification of neurons derived from the septal eminence, we used an intersectional approach based on *Ascl1* expression by fate-restricted neurogenic progenitors within the *Nkx2.1* lineage (*Kelly et al., 2018*; *Kelly et al., 2019*; *Figure 5A*). We combined the Nkx2.1-Flp driver mouse line with a tamoxifen (TMX)-inducible Ascl1-CreER^T2 line and the intersectional Ai65 reporter line, which expresses the fluorescent protein tdTomato in a Cre- and Flp-dependent manner (*Figure 5B*). Administration of tamoxifen causes the expression of tdTomato in cells with a developmental history of *Nkx2.1* expression that are undergoing a peak of *Ascl1* expression, leading to a neurogenic division (*Kelly et al., 2019*). We administered tamoxifen to timed-pregnant dams at four embryonic timepoints spanning the neurogenic period (E10, E12, E14, and E16) and collected the brains of their progeny at P30 when development of the septum is complete (*Figure 5C*). Temporally defined cohorts of tdTomato-expressing cells were distributed primarily within the LS (*Figure 5D*). Cells labeled at different stages were not only distributed in a general medial-to-lateral pattern, as expected (*Creps, 1974*), but also in a roughly medial-to-dorsal-to-ventral one. Considering the temporal dynamics and viewing each temporal cohort as a still picture within a sequence, this pattern can be compared to water in a fountain or a fireworks display, whereby cells are initially located in the center of an upward stream, 'moving' into more dorsal positions until they reach the apex and then 'falling on the outside' to progressively more ventral positions

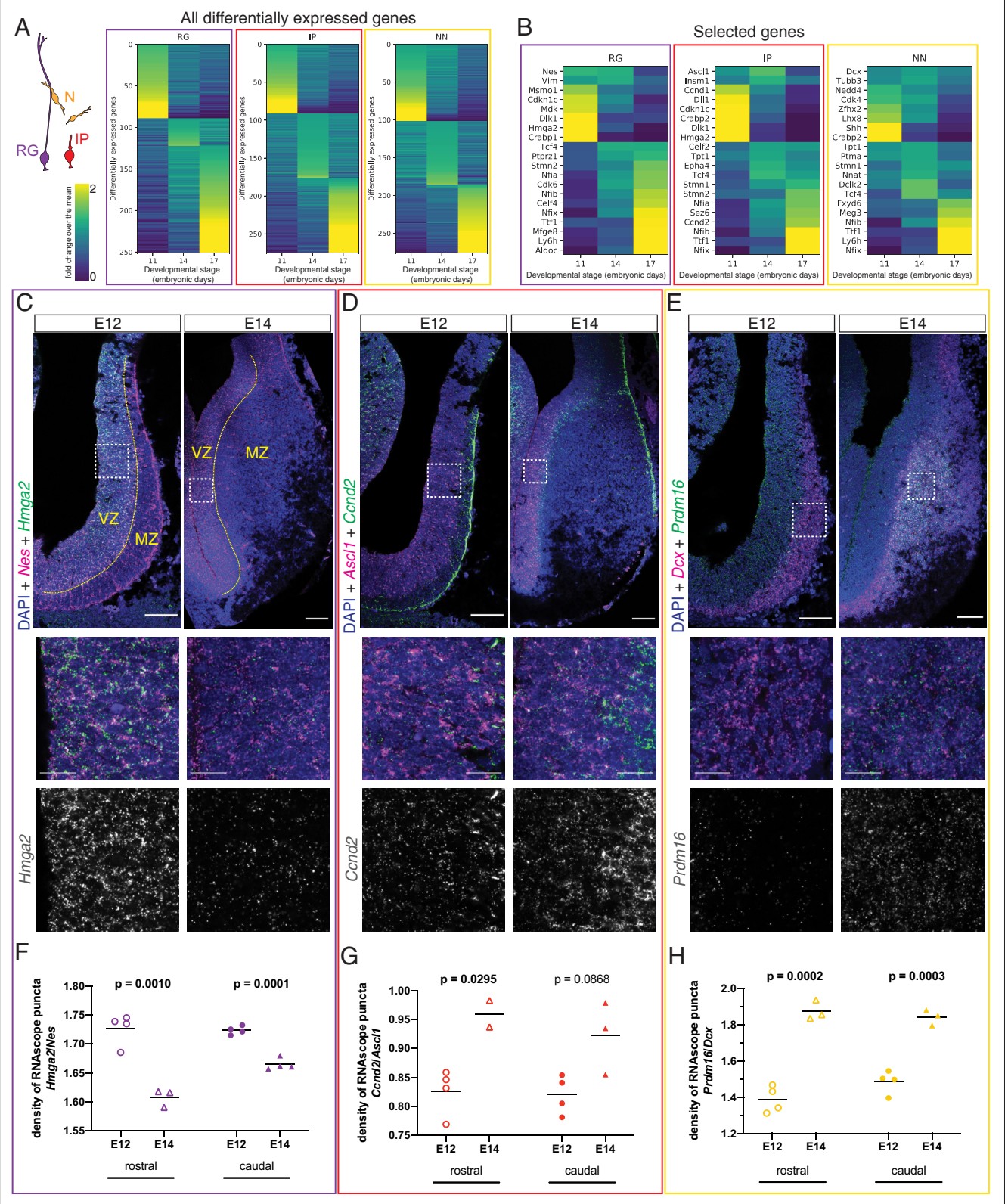

**Figure 4.** Different transcriptional programs are active during early and late septal neurogenesis. (**A**) Heatmaps illustrating all genes differentially enriched in the scRNA-Seq dataset across embryonic stages for the three cell types indicated in the cartoon: radial glia (RG, purple), intermediate progenitors (IP, red), and newborn neurons (NN, yellow). The Viridis color scale represents fold change over the mean, and applies to (**A**) and (**B**). (**B**) Heatmap showing the differential enrichment of a subset of selected genes, displayed as in (**A**). (**C–E**) Coronal brain sections showing the rostral septum

*Figure 4 continued on next page*

*Figure 4 continued*

at embryonic days (E)12 and 14, subjected to fluorescent single-molecule in situ hybridization for the genes *Nes+ Hmga2* (**C**), *Ascl1+ Ccnd2* (**D**), and *Dcx+ Prdm16* (**E**). Cell-type marker mRNA puncta (*Nes*, *Ascl1*, and *Dcx*) are displayed in magenta, mRNA puncta for differentially enriched genes (*Hmga2*, *Ccnd2*, and *Prdm16*) in green, and DAPI counterstaining in blue. Yellow dashed lines in (**C**) mark the limit between ventricular zone (VZ) and mantle zone (MZ), as indicated. Dashed boxes indicate the location of the magnified 100 × 100 µm fields shown below, both as merged images (middle panels) and single channel for the corresponding differentially enriched genes (bottom panels). Scale bars: 100 µm (top panels), 25 µm (middle panels). (**F–H**) Quantification of density of mRNA puncta of differentially enriched genes, normalized to the density of cell-type marker mRNA; measurements were obtained from the rostral (empty symbols) and caudal (full symbols) portions of the septum at E12 (circles) and E14 (triangles). All data points are represented; black bars represent the mean. Unpaired t-tests were performed; p-values are indicated above the corresponding compared sets of data: those highlighted in bold indicate statistically significant differences (p<0.05).

The online version of this article includes the following source data and figure supplement(s) for figure 4:

**Source data 1.** RNAscope puncta quantifications, *Figure 4*.

**Figure supplement 1.** Additional categories and examples of genes differentially enriched across developmental stages.

**Figure supplement 1—source data 1.** RNAscope puncta quantifications, *Figure 4—figure supplement 1*.

(*Figure 5D and E*, *Figure 5—figure supplement 1C*). This is consistent with a previous study where the authors proposed that specific neuronal subtypes might follow specific birthdate patterns divergent from the general medial-to-lateral order (*Wei et al., 2012*). We adopted and extended the classification of morphological types of LS neurons described by *Alonso and Frotscher, 1989* to determine the extent of morphological diversity within each neuronal temporal cohort. We found the same morphological neuronal types regardless of their location within the different nuclei within the LS, We therefore designated labeled cells as types I, II, and III, irrespective of their allocation to LSd, LSi, or LSv, as follows (see Materials and methods for further details): type I neurons, with relatively few thick dendrites forming a spherical contour; type II neurons, with thinner and branched dendrites; we propose to further subdivide the morphological types by adding a neuronal type III, with thick, spine-dense dendrites forming a bipolar dendritic field (*Figure 5F*). With the exception of E16, where only a few mostly type II cells were present in the LSv, there were neurons of all three types in each temporal cohort labeled, with slight changes in their overall proportions (*Figure 5G*). Type I cells were most abundant in the LSd; type II neurons were prevalent across all LS areas and temporal cohorts, and the proportion of type III cells was highest in the E14 cohort, and practically confined to LSd and LSi (*Figure 5—figure supplement 1A*). Type II neurons could be further classified into three subgroups based upon the morphology of their dendrites; type IIa, with overall thicker dendrites; type IIb, with a thick initial dendritic segment bifurcating into thinner processes, and type IIc, with thin and long dendrites (*Figure 5—figure supplement 1B*). The LSd/LSi location and bipolar shape of the dendritic field of type III neurons are likely reflective of unique functional/connectivity properties, which will require further investigation (*Figure 5—figure supplement 1C*). Taken together, our data suggest that while the output of morphological neuron types remains relatively stable over the course of neurogenesis, their allocation across LS nuclei is strongly associated with birthdate. This set of experiments confirms that while septal neurogenesis generally follows a medial-to-lateral organization, certain subsets of neurons, such as those from septal eminence progenitors, follow slightly different patterns (*Wei et al., 2012*). We expand upon the morphological classes proposed by Alonso and Frotscher, describing the newly defined type III neurons.

## Discussion

The full diversity of neurons contained within the septum and the dynamic transcriptional programs controlling their generation from embryonic progenitors is still largely unexplored. Foundational work from the Yang lab explored a number of neuronal subpopulations in the mature septum, demonstrating that they were derived from multiple progenitor regions located within the septum itself, the MGE/PoA, and the pallium or pallial/subpallial boundary (*Wei et al., 2012*). These findings were extended with the development of a septal-specific Zic4-Cre driver mouse line (*Rubin et al., 2010*), which allowed for a more refined molecular classification of septal progenitor zones. We propose to rename the caudal portion of the developing septum as septal eminence. This term refers to both its anatomical appearance, as an enlargement of the tissue towards the ventricular lumen caused by cellular proliferation, and to the main allocation of cells generated from this germinal zone in

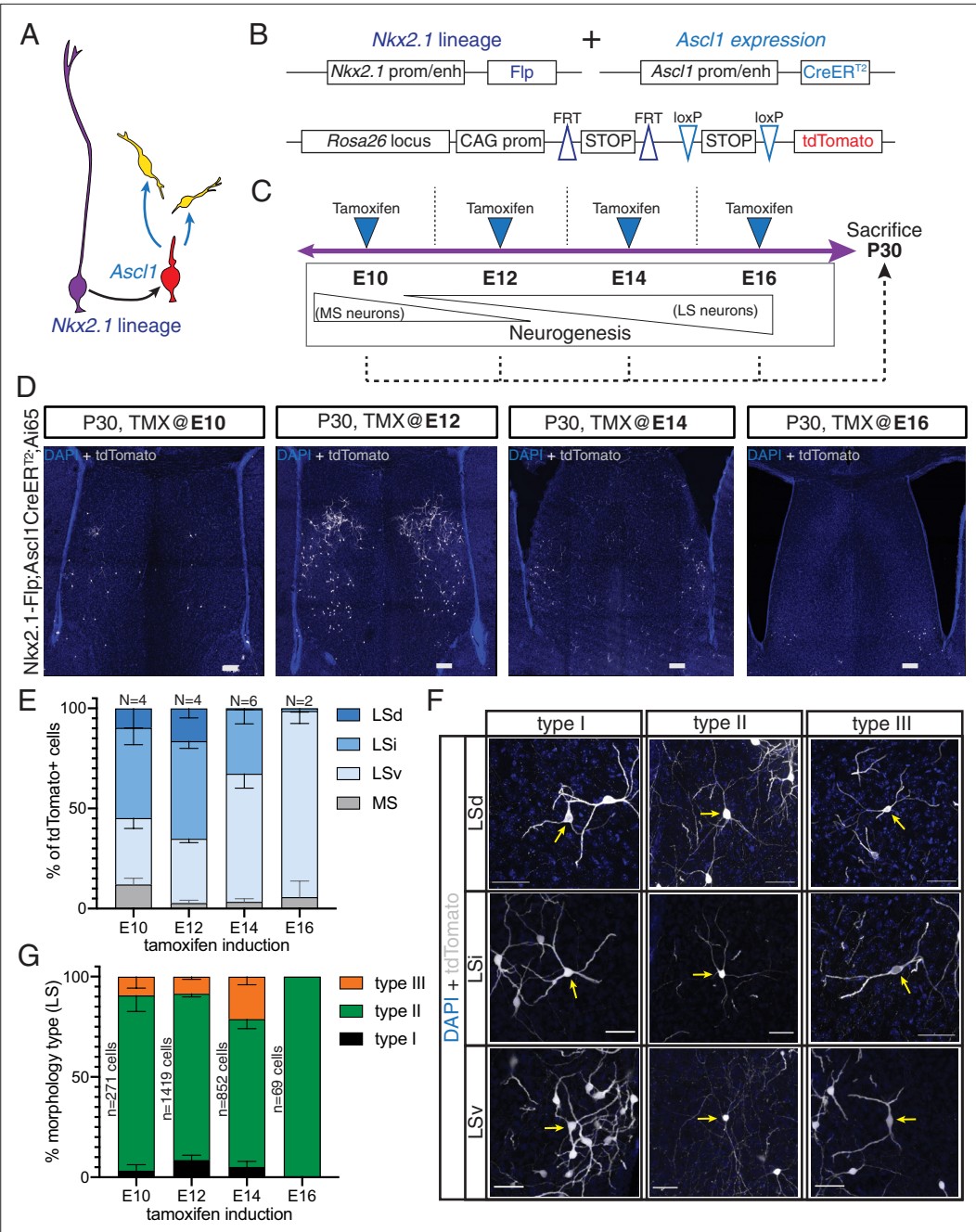

**Figure 5.** Generation of sequential temporal cohorts of neurons from the septal eminence. (**A**) Schematic of neurogenesis in the septal eminence: radial glia (RG) (purple) within the *Nkx2.1* lineage give rise to *Ascl1*-expressing transit-amplifying progenitors (red), which in turn divide to generate neurons. (**B**) Two driver lines, Nkx2.1-Flp and Ascl1-CreER[T2], were crossed with an intersectional reporter line. The action of both Flp and Cre recombinases (i.e., within the *Nkx2.1* lineage and in the presence of tamoxifen) leads to the expression of the fluorescent reporter tdTomato. (**C**) Experimental design: tamoxifen was administered to pregnant dams at embryonic day (E)10, E12, E14, or E16, covering the entire septal neurogenic period. Resulting litters were sacrificed and analyzed at postnatal day (P)30. (**D**) Representative images of coronal sections through the septum of P30 mice in which recombination was induced at the indicated stages. Cells derived from terminal progenitor divisions are labeled with tdTomato (gray; counterstained with DAPI, blue). Scale bars, 100 μm. (**E**) Quantification of the location of tdTomato+ cells in each septal nucleus as a percentage of total labeled cells (mean ± SD) within each temporal cohort. The number of biological replicates is indicated above each bar. (**F**) Representative images of neurons belonging to the three morphological subtypes within each LS nucleus, labeled with tdTomato (gray;

*Figure 5 continued on next page*

*Figure 5 continued*

counterstained with DAPI, blue). Arrows indicate the cell bodies of corresponding neuronal types. Scale bars, 50 μm. (**G**) Quantification of morphological neuron types as a percentage of the total number of classified cells (mean ± SD) within each temporal cohort. The total number of classified cells is indicated on the left side of the corresponding column; the number of biological replicates is the same as in (**E**) except for the E14 timepoint (N = 7).

The online version of this article includes the following source data and figure supplement(s) for figure 5:

**Source data 1.** Location and morphology of septal eminence derived temporal cohorts.

**Figure supplement 1.** Detailed distribution of morphological cell types across temporal cohorts of septal eminence-derived neurons.

**Figure supplement 1—source data 1.** Location and morphology of septal eminence derived temporal cohorts by septal nucleus; analysis of type II subtypes.

the mature brain. This region has long been recognized as a proliferative region, but for lack of a specific term it has been diversely referred to as 'the caudal part of the septum' (*Puelles et al., 2000*), the 'ventral septum' or 'posterior ventral septum' (*Hoch et al., 2015a*; *Hoch et al., 2015b*; *Rubin et al., 2010*; *Wei et al., 2012*), the 'subpallial septal neuroepithelium' (*Iyer and Tole, 2020*; *Magno et al., 2017*), or 'the most ventromedial part of the septum,' further defined by its combinatorial gene expression as 'pSe4' (*Flames et al., 2007*). The renaming we suggest is based on the similarities between the rostral and caudal portions of the developing septum and the LGE and the MGE progenitor zones, respectively. While the rostral septum and the LGE serve as the anatomical templates for their mature derivatives (the septum proper and the striatum), the MGE and the septal eminence are largely transient structures with merely vestigial counterparts in the mature brain (*Delgado and Lim, 2017*; *Delgado et al., 2020*; *Merkle et al., 2014*). The main role of the MGE and the septal eminence during embryonic development is to generate cells destined to occupy other areas of the brain (cortical interneurons in the MGE, multiple septal neurons and basal forebrain cholinergic neurons in the septal eminence). Additionally, both of these proliferative areas express *Nkx2.1* (*Marin et al., 2000*; *Puelles et al., 2000*; *Sussel et al., 1999*), a gene that is fundamental to maintain their correct regional identity and neuronal output.

Our data suggest that the proliferative areas of the septum proper and the septal eminence have diverging molecular identities, which likely lead to differences between the cell biology of the progenitors contained within them, in terms of both cell-type diversity and proliferative behavior. We observed that progenitors in the septal eminence divide in more apical locations than their rostral counterparts (*Figure 2D and E*), and that a higher proportion of ventricular divisions are ASCL1-positive (*Figure 2F and G*). This suggests that *Nkx2.1*-expressing septal eminence progenitors may undergo transit-amplifying cell divisions as short neural precursors rather than delaminated intermediate progenitors (*Petros et al., 2015*; *Turrero García and Harwell, 2017*). Future work comparing the specific cell biological features of septal progenitors located within defined subdomains should help to elucidate how cell fate specification occurs at the cellular and/or clonal level (*Zhou et al., 2020*).

Our work helps to clarify the contribution of the septal eminence to the diversity of neurons in the medial and lateral nuclei of the mature septum. Consistent with previous reports, we found that the vast majority of neurons throughout the LS, and about half of those in the medial septum, belong to the *Zic4* lineage (*Figure 3D and G*; *Magno et al., 2017*; *Rubin et al., 2010*). Roughly a quarter of those neurons have a developmental history of *Nkx2.1* expression (*Figure 3E and H*) and could thus originate from either the septal eminence or the MGE/PoA, as has been proposed before (*Wei et al., 2012*). To resolve this, we used an intersectional genetic approach to label cells expressing both *Nkx2.1* and *Zic4*, confirming that the vast majority of *Nkx2.1*-lineage neurons in the LS are derived from the septal eminence rather than from the MGE/PoA, since these regions do not express *Zic4* (*Magno et al., 2017*). A smaller pool of intersectional neurons was found in the MS (*Figure 3F*), raising the possibility that some of the MS neurons previously thought to be derived from the MGE (positive for ChAT, nNOS, CB, and/or PV) (*Wei et al., 2012*) could be generated in the septal eminence instead. The subtractive population, derived from the MGE/PoA, consisted of glia and a small proportion of neurons, both in the MS and in the LS (*Figure 3F*). While it is possible that these cells account for some of the cell types outlined above, further fate-mapping experiments should elucidate their molecular

identity and precise anatomical origin; for example, new genetic tools such as the Zic4-Cre driver line could be used to refine previous fate-mapping experiments performed with the Shh-Cre driver line (*Wei et al., 2012*), to address the possibility that neurons presumed to be PoA-derived might originate from the small portion of the vSe that expresses *Shh* (pSe6) (*Flames et al., 2007*). More work will be necessary to understand how the interplay of these and other genes with regionally restricted expression patterns, such as fibroblast growth factors (*Hoch et al., 2015a*), impacts the neuronal output of either septal progenitor zone. We hypothesize that further subdivisions within the developing septum (*Flames et al., 2007*) and/or different progenitor proliferation modes (*Turrero García and Harwell, 2017*) are responsible for the generation of specific neuronal subtypes, as in the MGE (*Hu et al., 2017*).

Temporal changes in the transcriptional profiles of ventral telencephalic progenitors guide their competence to generate different types of neurons as neurogenesis progresses (*Turrero García and Harwell, 2017*). We hypothesized that similar changes would underlie the inside-out patterning of the septum. To test this, we used our scRNA-Seq dataset to compare gene expression levels across developmental stages and cell types (*Figure 4*). Several of the genes we validated recapitulate temporal patterns of expression that have been described in other parts of the developing central nervous system. For example, *Hmga2* is highly expressed in early stages of neurogenesis across all septal cell types we analyzed (*Figure 4C and F*, *Figure 4—figure supplement 1B*); this is similar to cortical progenitors, where it controls early developmental programs (*Shu et al., 2019*). Likewise, we observed an increase in the level of expression of *Ccnd2* (*Figure 4D and G*) in intermediate progenitors at later developmental stages, which suggests that its role in cell cycle regulation and its interplay with *Ccnd1* (*Figure 4—figure supplement 1F*) are crucial for ensuring the correct neuronal outputs of late septal progenitors, as it is in the MGE (*Glickstein et al., 2007*; *Glickstein et al., 2009*). We also found a general upregulation in Nfi factors (*Figure 4B*, *Figure 4—figure supplement 1G*) during late stages of septal neurogenesis, reminiscent of their role in late fate specification in the retina (*Clark et al., 2019*). Further research will be necessary to determine whether these or other temporally enriched factors promote the specification of specific septal neuron fates. In this study, we have focused on potential fate determinants that show similar patterns of up- or downregulation in the septum proper and the septal eminence (*Figure 4F–H*, *Figure 4—figure supplement 1E–G*). Our dataset provides a springboard for exploring candidate factors downstream or in parallel to *Nkx2.1* that may be responsible for differences in progenitor composition and fate potential observed between the septal eminence and the septum proper. While location and time of progenitor divisions appear to determine the general identity of septal neurons (*Figure 1G and H*, *Figure 1—figure supplement 2D and E*, *Figure 2*, *Figure 3*), it is likely that the confluence of extrinsic factors and cell-cell interactions further refines their fate specification (*Fishell and Kepecs, 2020*).

Another aspect we considered when analyzing patterns of gene expression was the progression of neural progenitors from radial glia to neurons within each developmental stage. We noticed that *Prdm16* is highly expressed in both RG and newborn neurons, especially late-born (*Figure 4E and H*). In other parts of the developing brain, *Prdm16* is expressed exclusively in radial glia and quickly shut down as cells become intermediate progenitors (*Baizabal et al., 2018*; *Turrero García et al., 2020*). Its upregulation to even higher levels in septal neurons could reflect a novel role for this gene in the control of further aspects of neuronal differentiation, especially in the LS. Other transcription factors such as *Sp8*, *Pax6*, *Sox6*, or *Nkx2.1*, which are usually downregulated by postmitotic neurons in other parts of the brain, maintain high levels of expression in septal neurons into adulthood (*Magno et al., 2009*; *Wei et al., 2012*). While the role of several of these genes has been explored in postmitotic MGE-derived neurons (*Azim et al., 2009*; *Batista-Brito et al., 2009*; *Nóbrega-Pereira et al., 2008*), their continued expression in septal neurons has not been addressed so far. This phenomenon could be an intrinsic part of the process of septal neuron fate specification and/or maintenance of subtype-specific neuronal identity and circuitry (*Sheehan et al., 2004*).

Our analysis of differential gene expression across early and late neurogenic timepoints suggests that groups of neurons specified at the same time may share particular functional features. Earlier studies have analyzed the birthdates of mature septal neurons expressing specific markers, and thus assumed to share a common origin (*Wei et al., 2012*). Here, we used a genetic strategy to label

isochronic temporal cohorts of septal eminence-derived neurons (*Figure 5A–C*). We found that cells within the *Nkx2.1* lineage are largely located in the LS and generated in a dorsal-to-ventral, rather than medial-to-lateral, gradient (*Figure 5C and D*). Our labeling strategy allowed us to study the morphology of these neurons (*Figure 5E*) and the distribution of defined morphological subtypes (*Figure 5F*, *Figure 5—figure supplement 1A*). The allocation of *Nkx2.1*-lineage neurons across the three nuclei of the LS varied dramatically over the course of neurogenesis, producing the fountain-like pattern of temporal cohorts (*Figure 5E*). In contrast, the proportion of the three morphological types produced by the septal eminence remained relatively stable throughout the neurogenic period, save for the exclusive production of a small population of type II neurons at the close of the neurogenic period (*Figure 5G*). The relative stability of morphological types across LSd, LSi, and LSv, despite the differences in connectivity and cell-type composition among these areas (*Risold and Swanson, 1997a*; *Risold and Swanson, 1997b*; *Swanson and Cowan, 1979*), suggests that the temporally defined gene expression programs we observe impart positional rather than morphological constrictions onto each *Nkx2.1*-lineage temporal cohort. The final position that each cohort occupies as its neurons mature would entail a unique combination of possible inputs and outputs to further refine their identity and thus their function, including their intrinsic electrophysiology, connectivity, and ultimately their role in behavioral regulation (*Wang et al., 2019*). More fine-grain studies addressing the extent of septal neuron diversity, within this and other lineages, will be necessary to obtain a complete picture of how developmental origin and maturation processes work in concert to establish the mature circuitry between the septum and the rest of the brain.

We based our analyses in the foundational work of *Alonso and Frotscher, 1989*, with three additions: (1) since the morphological classes within the *Nkx2.1* lineage were not clearly segregated across LSd and LSi (*Figure 5—figure supplement 1A*), we decided to unify the nomenclature, basing it in the type I and type II categories described for the LSd and extending it to the LSv as well; (2) we consistently observed neurons with thick dendrites and bipolar shape, which we designated as type III septal neurons (*Figure 5E*); and (3) given the morphological variability within type II cells, we propose to further classify them into subtypes IIa, IIb, and IIc (*Figure 5—figure supplement 1B*). Previous studies, confirmed by our own fate-mapping experiments, have described the presence of numerous *Nkx2.1*-lineage cells in the medial septum (*Figure 3B and C*), including most cholinergic MS neurons (*Magno et al., 2017*) as well as a high proportion of astrocytes (*Figure 3—figure supplement 1*). However, in our temporal cohort experiments only a small fraction of labeled cells were located in the MS, and we did not observe cells with obvious glial morphology in either MS or LS (*Figure 5D–F*, *Figure 5—figure supplement 1B*). This hints at the possibility that there are different specification programs for distinct cell types within the *Nkx2.1* lineage, where some cell types such as MS cholinergic neurons and astrocytes could be generated by direct neurogenic divisions of septal and/or MGE/PoA progenitors, bypassing an *Ascl1*-expressing transit-amplifying state that appears to be prevalent when LS neurons are generated (*Figure 5A*). One important caveat of our current approach is that we have not been able to correlate morphological types to transcriptional profiles, beyond the fact that they would belong to an LS group (clusters 2, 5, and 7 in *Figure 1—figure supplement 3A*). This is due in great part to the relative scarcity of our mature septum dataset and should be addressed by future experiments where a higher resolution can be achieved. It will be important to assess if and how these morphological types differ in terms of their molecular identity and connectivity patterns in order to better understand their function within septal circuits and consequently their role in behavioral regulation. It will be particularly interesting to study how the three morphological types we have observed might be correlated to the three classes of LS neurons that have been described based on their electrophysiological properties (*Gallagher et al., 1995*; *Wang et al., 2019*).

Single-cell sequencing techniques allow unprecedented interrogation of the molecular diversity of cell types present in any tissue. Here, we provide the first scRNA-Seq dataset of the developing and mature septum that we are aware of, in a format that will allow other investigators to use it as a springboard towards further discoveries. Since we included the MGE/PoA in our dissections, our data can be used to complement and extend previously published scRNA-Seq datasets addressing cell diversity in this area (*Mayer et al., 2018*; *Mi et al., 2018*). Our current analysis highlights the point in the molecular trajectory of septal progenitors when they acquire distinct states that are predictive

of their cardinal neuronal subtype identity (*Figure 1G and H*). Our data suggest that the mechanism for determination of cardinal cell-type identity is similar to what is observed in neighboring structures such as at the MGE (*Fishell and Kepecs, 2020*).

The initial exploration of P30 samples within our dataset has yielded several previously undescribed markers of neuronal subpopulations in the adult septum (*Figure 1—figure supplement 3A–C*), one of which we validated with a transgenic mouse line that grants access to GABAergic neurons in the LS (*Figure 1—figure supplement 3D and E*). However, this study focuses largely on developmental stages; future research efforts should address mature septal neuronal diversity in a more systematic and comprehensive way, ideally correlating mature neuronal types with their developmental origin to complement this and previous studies (*Wei et al., 2012*). The septum is involved in numerous psychological and psychiatric conditions, including psychotic spectrum disorders, anxiety, and depression (*Sheehan et al., 2004*), Despite this, very few detailed descriptions of the human septum (*Andy and Stephan, 1968*) or its development (*Brown, 1983*; *Rakic and Yakovlev, 1968*) have been published. Considering the high evolutionary conservation of septal nuclei across tetrapods in terms of both anatomy and function (*Lanuza and Martínez-García, 2009*), studies like ours are likely to shed light on common mechanisms of cell fate determination and uncover species-specific cell types and developmental programs. More detailed molecular comparisons across multiple species, including humans, will be necessary to fully understand how septal neuronal diversity is specified during development, and how it impacts brain function and behavior.

# Materials and methods

**Key resources table**

| Reagent type (species) or resource | Designation | Source or reference | Identifiers | Additional information |
|---|---|---|---|---|
| Antibody | Mouse monoclonal anti-ASCL1 | BD Pharmingen | Cat# 556604; RRID:AB_396479 | (1:100) |
| Antibody | Chicken polyclonal anti-GFP | Aves | Cat# GFP-1020; RRID:AB_10000240 | (1:1000) |
| Antibody | Rat monoclonal anti-pH3 | Abcam | Cat # ab10543; RRID:AB_2295065 | (1:500) |
| Antibody | Mouse monoclonal anti-NeuN | Millipore | Cat# MAB377; RRID:AB_2298772 | (1:500) |
| Antibody | Rabbit polyclonal anti-NKX2.1 | Santa Cruz | Cat# sc-53136; RRID:AB_793529 | (1:250) |
| Antibody | Chicken polyclonal anti-RFP | Rockland | Cat# 600-901-379; RRID:AB_10704808 | (1:1000) |
| Antibody | Rabbit polyclonal anti-RFP | Rockland | Cat# 600-401-379; RRID:AB_2209751 | (1:1000) |
| Antibody | Rabbit polyclonal anti-ZIC | Segal Lab, DFCI; *Borghesani et al., 2002* | n/a (gift) | (1:500) |
| Antibody | Goat polyclonal anti-chicken Alexa 488 | Thermo Fisher | Cat# A11039; RRID:AB_142924 | (1:1000) |
| Antibody | Goat polyclonal anti-chicken Alexa 546 | Thermo Fisher | Cat# A11040; RRID:AB_1500590 | (1:1000) |
| Antibody | Goat polyclonal anti-mouse Alexa 488 | Thermo Fisher | Cat# A11001; RRID:AB_2534069 | (1:1000) |
| Antibody | Goat polyclonal anti-rabbit Alexa 488 | Thermo Fisher | Cat# A11008; RRID:AB_143165 | (1:1000) |
| Antibody | Goat polyclonal anti-rabbit Alexa 546 | Thermo Fisher | Cat# A11010; RRID:AB_2534077 | (1:1000) |
| Antibody | Goat polyclonal anti-rat Alexa 647 | Thermo Fisher | Cat# A21247; RRID:AB_141778 | (1:1000) |
| Strain, strain background (*Mus musculus*) | B6.Cg-*Gt(ROSA)26Sor*^tm14(CAG-tdTomato)Hze/J ('Ai14' in text) | Jackson Laboratory | Stock no. 007914; RRID:IMSR_JAX:007914 | (*Gt(ROSA)26Sor*) |

*Continued on next page*

*Continued*

| Reagent type (species) or resource | Designation | Source or reference | Identifiers | Additional information |
|---|---|---|---|---|
| Strain, strain background (*M. musculus*) | B6;129S-*Gt(ROSA)26Sor*<sup>tm65.1(CAG-tdTomato)Hze</sup>/J ('Ai65' in text) | Jackson Laboratory | Stock no. 021875; RID:IMSR_JAX:021875 | (*Gt(ROSA)26Sor*) |
| Strain, strain background (*M. musculus*) | STOCK *Ascl1*<sup>tm1.1(Cre/ERT2)Jejo</sup>/J ('Ascl1-CreER<sup>T2</sup>' in text) | Jackson Laboratory | Stock no. 012882; RRID:IMSR_JAX:012882 | (*Ascl1*) |
| Strain, strain background (*M. musculus*) | CD-1 ('wildtype; WT' in text) | Charles River | Strain code 022; RRID:IMSR_CRL:022 | |
| Strain, strain background (*M. musculus*) | B6.Cg-*Gt(ROSA)26Sor*<sup>tm1.3(CAG-tdTomato,-EGFP)Pjen</sup>/J ('FLTG' in text) | Jackson Laboratory | Stock no. 026932; RRID:IMSR_JAX:026932 | (*Gt(ROSA)26Sor*) |
| Strain, strain background (*M. musculus*) | C57BL/6J-Tg(Nkx2-1-cre)2Sand/J ('Nkx2.1-Cre' in text) | Jackson Laboratory | Stock no. 008661; RRID:IMSR_JAX:008661 | (*Nkx2-1*) |
| Strain, strain background (*M. musculus*) | *Nkx2-1*<sup>tm2.1(flpo)Zjh</sup>/J ('Nkx2.1-Flp' in text) | Jackson Laboratory | Stock no. 028577; RRID:IMSR_JAX:028577 | (*Nkx2-1*) |
| Strain, strain background (*M. musculus*) | STOCK Tg(*Prkcd-glc-1/CFP,cre*)<sup>EH124Gsat</sup>/Mmucd ('Prkcd-Cre' in text) | Jackson Laboratory | MMRRC:011559; RRID:MMRRC_011559-UCD | (*Prkcd*) |
| Strain, strain background (*M. musculus*) | B6;129-*Gt(ROSA)26Sor*<sup>tm5(CAG-Sun1/sfGFP)Nat</sup>/J ('Sun1-GFP' in text) | Jackson Laboratory | Stock no. 021039; RRID:IMSR_JAX:021039 | (*Gt(ROSA)26Sor*) (*Sun1*) |
| Strain, strain background (*M. musculus*) | Zic4-iCre ('Zic4-Cre' in text) | Kessaris Lab, UCL (*Rubin et al., 2010*) | Animal code A611 (gift) | (*Zic4*) |
| Other | *Ascl1* RNAscope probe | ACD | Cat# 313291 | |
| Other | *Ccnd1* RNAscope probe | ACD | Cat# 442671 | |
| Other | *Ccnd2* RNAscope probe | ACD | Cat# 433211 | |
| Other | *Dcx* RNAscope probe | ACD | Cat# 478671 | |
| Other | *Hes5* RNAscope probe | ACD | Cat# 400998 | |
| Other | *Hmga2* RNAscope probe | ACD | Cat# 466641 | |
| Other | *Nes* RNAscope probe | ACD | Cat# 313161 | |
| Other | *Nfia* RNAscope probe | ACD | Cat# 586501 | |
| Other | *Prdm16* RNAscope probe | ACD | Cat# 584281 | |
| Software, algorithm | Fiji 2.1.0/1.53c | *Schindelin et al., 2012* | http://fiji.sc; RRID:SCR_002285 | |
| Software, algorithm | Prism 9 | GraphPad | RRID:SCR_002798 | |
| Software, algorithm | MATLAB (MATBOTS) | MathWorks | RRID:SCR_001622 | |

## Experimental model and subject details

All animal procedures conducted in this study followed experimental protocols approved by the Institutional Animal Care and Use Committee of Harvard Medical School. Mouse strains mentioned in the main text are listed in the Key resources table. Wild-type animals for single-cell sequencing and validation experiments were purchased as timed-pregnant females or as entire litters at the following developmental timepoints: E11, 14, and 17; and P3, 10, and 30. For all other experiments, mouse housing and husbandry were performed in accordance with the standards of the Harvard Medical School Center of Comparative Medicine. Mice were group housed in a 12 hr light/dark cycle, with access to

food and water ad libitum. Samples were obtained at the ages indicated in the figure legends and throughout the text; for embryonic samples, the plug date was considered as E0. All results reported include animals of both sexes, balanced equally wherever possible; the sex of embryos and P3 animals was not determined. The number of animals used for each experiment (i.e., biological replicates) is indicated in the corresponding graphs where possible. The number of animals used for scRNA-Seq experiments was (the number of cells that passed the initial quality controls for each stage is indicated in brackets): E11 – 35 embryos from three litters (25,829 cells); E14 – 33 embryos from three litters (19,555 cells); E17 – 8 embryos from two litters (10,526 cells); P3 – 4 pups from two litters (7355 cells); P10 – 2 pups (one male, one female) from one litter (4833 cells); P30 – 2 males and 2 females from one litter (6456 cells).

## Method details

### Dissection, single-cell suspension and droplet capture

For embryonic samples, the pregnant dam was sacrificed; embryos were removed from the uterus and maintained in Hibernate-E medium minus CaCl$_2$ (HEMC), on ice for the remainder of the procedure. Postnatal animals were deeply anesthetized and transcardially perfused with ice-cold PBS. Brains were extracted and transferred to dissection medium (HEMC +0.1 mg/ml DNAse I). All septa (MGEs were also collected from embryonic samples) were manually dissected out and lightly minced with the dissection forceps, then transferred with a minimum amount of medium to an Eppendorf tube containing 1 ml of Accutase and 0.1 mg/ml of DNAse I. They were then incubated, rocking at 4°C, for 10–15 min. The tube was then centrifuged at 1000 rpm, at 4°C, for 1–2 min. After discarding the supernatant, the tissue was resuspended in 1 ml of dissection medium and dissociated by gently pipetting up and down, first with a 1000 µl pipette tip (10–15 times), and subsequently with a 200 µl tip (10 times), both loaded to half to two-thirds capacity. The suspension was centrifuged at 1000 rpm, at 4°C, for 5 min. The supernatant was discarded, and cells were resuspended in 1 ml of HEMC and filtered through a 35 µm cell strainer and transferred to a clean low-adhesion Eppendorf tube. The resulting single-cell suspension was maintained on ice and subjected to single-cell droplet encapsulation with a custom microfluidic inDrops system at the Harvard Medical School Single Cell Core. Cell encapsulation and library preparation followed a previously described protocol (*Klein et al., 2015*; *Zilionis et al., 2017*), with modifications in the primer sequences as included in the Key resources table.

### Single-cell RNA sequencing

Libraries of approximately 3000 cells were collected from each sample. inDrops was performed as previously described (*Hrvatin et al., 2018*; *Klein et al., 2015*; *Zilionis et al., 2017*), generating indexed libraries that were then pooled and sequenced across eight runs on the NextSeq 500 (Illumina) platform.

### inDrops sequencing data processing

Transcripts were processed according to a previously published pipeline (*Hrvatin et al., 2018*; *Klein et al., 2015*; *Zilionis et al., 2017*). Briefly, this pipeline was used to build a custom transcriptome from the Ensembl GRCm38 genome and **GRCm38.88** annotation with Bowtie 1.1.1 (after filtering the annotation gtf file (ftp://ftp.ensembl.org/pub/release-88/gtf/mus_musculus/Mus_musculus.GRCm38.88.gtf.gz filtered for feature_type = 'gene', gene_type = 'protein_coding' and gene_status = 'KNOWN')). Read quality control and mapping against this transcriptome were then performed. Finally, unique molecular identifiers were used to reference sequence reads back to individual captured molecules, thus yielding values denoted as UMI counts. All steps of the pipeline were run with default parameters unless explicitly specified.

### Quality control for cell inclusion

Cells from each timepoint (E11, E14, E17, P3, P10, P30) were preprocessed separately. Any cells with fewer than 700 or more than 10,000 transcript counts were excluded from the analysis. Any cells in which >50% of UMIs mapped to mitochondrial genes were excluded. The dataset was normalized (NormalizeData()), variable genes identified (FindVariableGenes(x.low.cutoff = 0.0125, x.high.cutoff = 3, y.cutoff = 0.5)). The data were scaled using variable genes and a negative binomial model with

the percentage of mitochondrial genes and the number of UMIs per cell regressed (ScaleData(vars.to. regress = c("percent.mito", "nUMI"), genes.use = seurat_mat@var.genes, model.use = "negbinom")). PCA analysis, clustering, t-SNE plotting, and marker identification were performed using recommended parameters: RunPCA(pc.genes = seurat_mat@var.genes, pcs.compute = 40, pcs.print = 1:30, maxit = 500, weight.by.var = FALSE); FindClusters(dims.use = 1:30, resolution = 1.5, print.output = 1, save.SNN = T,reduction.type = "pca"); RunTSNE(dims.use = 1:30, do.fast = T); FindAllMarkers(only. pos = F, min.pct = 0.1, thresh.use = 0.25). By inspection of the t-SNE plots and marker genes, 1–3 clusters were identified at each timepoint as likely doublet clusters and those cells were excluded from further analysis. Our dataset after quality control contained 72,243 cells with more than 700 reads assigned to each cell.

## Dimensionality reduction and clustering

All 72,243 cells were combined into a single dataset and analyzed simultaneously. The R software package Seurat (*Butler et al., 2018*; *Satija et al., 2015*) was used to cluster cells. First, the data were log-normalized and scaled to 10,000 transcripts per cell. Variable genes were identified using the FindVariableGenes() function. The following parameters were used to set the minimum and maximum average expression and the minimum dispersion: x.low.cutoff = 0.0125, x.high.cutoff = 3, y.cutoff = 0.5. Next, the data were scaled using the ScaleData() function, and principal component analysis (PCA) was carried out. The FindClusters() function using the top 30 principal components (PCs) and a resolution of 1.5 was used to determine the initial 32 clusters.

## SPRING

SPRING plots were generated using the standard SPRING pipeline (*Weinreb et al., 2018*) with modifications described in *Weinreb et al., 2020*. Briefly, UNI counts were total counts normalized (without log-normalization) and filtered for highly variable genes. Gene expression values were standardized to zero mean and unit variance, and a low-dimensional embedding was estimated with PCA. The final 2D layout was produced by applying the ForceAtlas2 graph layout algorithm to a k-nearest-neighbor graph over PCA coordinates with k = 3.

The SPRING plots described in the paper and their data can be visualized and explored in the following webpages:

> *Figure 1C*, *Figure 1—figure supplement 2A–C* (all cells): https://kleintools.hms.harvard.edu/ tools/springViewer_1_6_dev.html?cgi-bin/client_datasets/Turrero_Garcia_et_al_2021_Mouse_ septum_development/all_cells
> *Figure 1D* (neuronal lineage): https://kleintools.hms.harvard.edu/tools/springViewer_1_6_ dev.html?cgi-bin/client_datasets/Turrero_Garcia_et_al_2021_Mouse_septum_development/ neuronal_lineage
> *Figure 1E and G*, *Figure 1—figure supplement 2D* (E11): https://kleintools.hms.harvard.edu/ tools/springViewer_1_6_dev.html?cgi-bin/client_datasets/Turrero_Garcia_et_al_2021_Mouse_ septum_development/E11
> *Figure 1F and H*, *Figure 1—figure supplement 2E*, *Figure 2A* (E14): https://kleintools.hms. harvard.edu/tools/springViewer_1_6_dev.html?cgi-bin/client_datasets/Turrero_Garcia_et_al_ 2021_Mouse_septum_development/E14

## Bioinformatic analyses: cluster ID and differential gene expression

Clusters were assigned a cell-type label by manual inspection of marker gene expression. We curated a list of known markers from the literature and constructed as cluster-by-marker heatmap as follows: UMI counts for each cell were total counts normalized (no log-normalization). The normalized counts were used to compute an average gene expression level for each marker in each cluster. The cluster averages for each gene were then standardized to zero mean and unit variance for visualization on a common scale.

Differential expression across timepoints was performed separately for each cell type using the 'rank_genes_groups' function in scanpy (*Wolf et al., 2018*). We followed the recommended preprocessing and used default parameters: cells were total counts normalized and then log transformed with pseudocount 1. A t-test was used to test significance with Benjamini–Hochberg correction for

multiple hypotheses. Heatmaps for differentially expressed genes report the degree of enrichment as fold change over the average expression across timepoints.

## Tamoxifen administration

For temporal cohort analyses (*Figure 5*), pregnant dams were administered 1–3 mg of tamoxifen (stock solution 10 mg/ml in corn oil) via oral gavage, at the corresponding embryonic stage.

## Tissue processing for immunofluorescence staining and fluorescent in situ hybridization (FISH)

Postnatal animals were transcardially perfused with PBS followed by 4% paraformaldehyde (PFA) in 120 mM phosphate buffer; their brains were dissected out and postfixed in 4% PFA overnight at 4°C. Brains were sectioned into 75–100 µm sections on a vibratome, and either further processed for FISH and/or immunofluorescence staining or stored at 4°C in PBS with 0.05% sodium azide. Embryonic brains were dissected out in ice-cold PBS and fixed in 4% PFA overnight at 4°C. The samples were cryoprotected in 30% sucrose/PBS overnight at 4°C, embedded in O.C.T. compound, frozen on dry ice, and stored at –20°C. Samples were sectioned at 20 µm on a cryostat; sections were either stored at –20°C or further processed for FISH and/or immunofluorescence staining.

## Immunofluorescence staining

### Floating vibratome sections

Samples were permeabilized with 0.5% Triton X-100 in PBS for 1–2 hr and blocked in blocking buffer (10% goat serum, 0.1% Triton X-100 in PBS) for 1–2 hr at room temperature. The sections were then incubated for 24–72 hr, at 4°C, with primary antibodies diluted in blocking buffer. The samples were washed three times (10–30 min/wash) with PBS, counterstained with 4′,6-diamidino-phenylindole (DAPI) for 45 min (both steps at room temperature), and incubated with secondary antibodies diluted in blocking buffer for 2 hr at room temperature or overnight at 4°C. They were then washed (three 10–30 min washes) and mounted on slides with ProLong Gold Antifade Mountant.

### Cryosections

Slides were allowed to reach room temperature, and then washed three times with PBS. Sections were permeabilized with 0.5% Triton X-100 in PBS for 30 min and blocked with blocking buffer for 1 hr at room temperature. Slides were incubated with primary antibodies diluted in blocking buffer overnight in a humid chamber at 4°C. They were then washed with PBS (three 10–30 min washes), counterstained with DAPI (45 min), and incubated for 1–2 hr with secondary antibodies diluted in blocking buffer, at room temperature. Slides were washed (three 10–30 min washes) with PBS and mounted with ProLong Gold Antifade Mountant.

### FISH

Embryonic samples at E12 and E14, prepared as outlined above, were submitted to the RNAscope protocol (Advanced Cell Diagnostics), following the manufacturer's instructions with minor modifications. All RNAscope probes (as listed in the Key resources table) were purchased from ACD. Additional in situ images were obtained from the Allen Developing Mouse Brain Atlas (*Figure 1G and H*, *Figure 1—figure supplement 2*) and from the Allen Mouse Brain Atlas (*Lein et al., 2007*; *Figure 1— figure supplement 3B and C*).

## Microscopy and image analysis

Images were acquired using a Leica SP8 laser point scanning confocal microscope. 10×, 25×, and 40× objectives were used, and the parameters of image acquisition (speed, resolution, averaging, zoom, z-stack, etc.) were adjusted for each set of samples. Images were further analyzed using ImageJ, both in its native and Fiji distributions, as described below. Brightness and contrast were adjusted as necessary for visualization; the source images were kept unmodified.

## Quantification and statistical analysis

### Cell quantification

The CellCounter tool in ImageJ/Fiji was used for all cell quantifications. In the mature septum (*Figures 3 and 5*), all cells positive for the corresponding marker within the dorsal, intermediate, and ventral nuclei of the LS (labeled in graphs as LSd, LSi, and LSv, respectively), as well as in the medial septal nucleus (MS) were counted. In *Figure 3D–F* and *Figure 3—figure supplement 1A, D and E*, the rostral, medial, and caudal (R/M/C) locations correspond approximately to Bregma +0.75, +0.5, and +0.25, respectively.

Cell morphology types (*Figure 5*, *Figure 5—figure supplement 1*) were determined based on the previous classification by *Alonso and Frotscher, 1989*. Examples are provided in *Figure 5F*. The key descriptive criteria for each neuronal type are:

- Type I neurons have relatively few (3–5) thick and sparsely ramified dendrites, with numerous spines, that form a small, roughly spherical dendritic field surrounding a round or oval soma.
- Type II neurons have thinner and branched dendrites of variable length and orientation and fewer spines, and a round or triangular cell body.
- Type III neurons have thick, spine-dense dendrites that form a bipolar dendritic field from an oval cell body with a characteristic spindle shape.

For quantification of dividing cells in the rostral and caudal portions of the developing septum (*Figure 2*), the septal eminence was identified either by the expression of the tdTomato fluorescent reporter in *Nkx2.1*-expressing cells and their progeny (*Figure 2D and E*) or by immunostaining for NKX2.1 itself (*Figure 2F and G*). In the former case, dividing (late G2/M) cells were identified by pH3 staining; in the latter, cells in M phase (late prophase to late telophase/cytokinesis) were identified from the DAPI counterstaining.

Quantification of RNAscope puncta (*Figure 4*, *Figure 4—figure supplement 1*) was performed using an automated data processing pipeline in MATLAB, guided by the SpotsInNucleiBot (https://hms-idac.github.io/MatBots (*Cicconet, 2018*). Each data point corresponds to the average values from the analysis of six fields (dimensions: 100 × 100 µm), located at ventral, intermediate, and dorsal positions within the embryonic septum on both hemispheres of a single embryo, analyzed at two different levels along the rostrocaudal axis as indicated. Fields for analyses were obtained from the areas within each sample that had highest levels of expression for the corresponding cell-type marker genes (RG – *Nes*; IP – *Ascl1*; NN – *Dcx*), as indicated in *Figure 4C–E*. Values are presented as the density of puncta for each mRNA analyzed normalized to the density of puncta for the corresponding cell-type marker.

Cell and RNA puncta numbers were compiled in Microsoft Excel spreadsheets; GraphPad Prism 9 was used to build graphs.

### Statistical analysis

All statistical analyses were performed with GraphPad Prism 9, as detailed in the figure legends. All p-values were rounded to ten thousandth and are presented above each statistical comparison in the corresponding figures; those highlighted in bold are below 0.05, which was considered the cutoff for statistical significance (p-values deemed not statistically significant under this criterion are displayed in regular type).

## Acknowledgements

The authors sincerely thank Alex Ratner and Mandovi Chatterjee at the Single Cell Core at Harvard Medical School for performing the inDrops runs and providing experimental advice; Nicoletta Kessaris, Z Josh Huang, Lucas Cheadle, and Michael Greenberg for providing mice; Maria Pazyra and Rosalind Segal for their gift of anti-ZIC antibody; Allon Klein for advice and support, including hosting the SPRING dataset; Harwell, Goodrich, Lehtinen, and Segal lab members for discussions and feedback; and Gord Fishell, Lisa Goodrich, Alex Pollen, and Matthew Schmitz for their comments on the manuscript.

# Additional information

## Funding

| Funder | Grant reference number | Author |
| --- | --- | --- |
| National Institute of Mental Health | R01MH119156 | Corey C Harwell |
| National Institute of Neurological Disorders and Stroke | R01NS102228 | Corey C Harwell |
| Ellen and Melvin Gordon Center for the Cure of Paralysis | Fellowship | Miguel Turrero García |
| Boehringer Ingelheim Fonds | MD Fellowship | Sarah K Stegmann |
| Bill and Melinda Gates Foundation | Millennium Scholarship | Tiara E Lacey |
| Howard Hughes Medical Institute | Gilliam Fellowship for Advanced Study | Christopher M Reid |
| Harvard Brain Science Initiative | Seed Grant | Corey C Harwell |
| Giovanni Armenise-Harvard Foundation | Junior Faculty Award | Corey C Harwell |

The funders had no role in study design, data collection and interpretation, or the decision to submit the work for publication.

## Author contributions

Miguel Turrero García, Conceptualization, Investigation, Supervision, Visualization, Writing – original draft, Writing – review and editing; Sarah K Stegmann, Tiara E Lacey, Christopher M Reid, Investigation, Writing – review and editing; Sinisa Hrvatin, Manal A Adam, Data curation, Formal analysis, Visualization; Caleb Weinreb, Data curation, Formal analysis, Visualization, Writing – review and editing; M Aurel Nagy, Formal analysis; Corey C Harwell, Conceptualization, Funding acquisition, Resources, Supervision, Writing – original draft, Writing – review and editing

## Author ORCIDs

Miguel Turrero García (iD) http://orcid.org/0000-0002-7294-169X
Corey C Harwell (iD) http://orcid.org/0000-0002-8043-5869

## Ethics

All animal procedures conducted in this study followed experimental protocols approved by the Institutional Animal Care and Use Committee of Harvard Medical School (IS961-3 and IS677-3).

## Decision letter and Author response

Decision letter https://doi.org/10.7554/eLife.71545.sa1
Author response https://doi.org/10.7554/eLife.71545.sa2

# Additional files

## Supplementary files

• Transparent reporting form

## Data availability

Sequencing data have been deposited in GEO under accession code GSE184879.

The following dataset was generated:

| Author(s) | Year | Dataset title | Dataset URL | Database and Identifier |
|---|---|---|---|---|
| Garcia MT, Hrvatin S, Nagy MA, Harwell CC | 2021 | Data from: Transcriptional profiling of sequentially generated septal neuron fates | http://www.ncbi.nlm.nih.gov/geo/query/acc.cgi?acc=GSE184879 | NCBI Gene Expression Omnibus, GSE184879 |

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
