## [Editor Report]

This paper captures the developmental trajectories of septal neurons and unravels their genetic codes. The authors use both single cell sequencing and transgenic mice to quantify septal cells derived from two septal progenitor zone, assessing the temporal dynamics of septal cells generated from one of these zones and classify the neurons according to morphology. The work will be of interest to developmental neurobiologists studying forebrain neurogenesis. The work is novel and provides substantial molecular insight into transcriptomic profiles of cells in the septum. The work will serve as a reference for future brain networks analysis.

---

## [Decision Letter]

**Decision letter after peer review:**

Thank you for submitting your article "Transcriptional profiling of sequentially generated septal neuron fates" for consideration by *eLife*. Your article has been reviewed by 2 peer reviewers, and the evaluation has been overseen by Joseph Gleeson as the Reviewing Editor and Marianne Bronner as the Senior Editor. The reviewers have opted to remain anonymous.

The reviewers have discussed their reviews with one another, and the Reviewing Editor has drafted this to help you prepare a revised submission. The reviewers were overall positive in their assessment, but Reviewer 1 has questions about the identity of some of the cell types, and also suggests modifications and corrections on the anatomical septal boundaries. Reviewer 2 has fewer direct comments, but was less enthusiastic about the impact of the paper without comparative analysis with a second species. Based upon the level of criticism for Reviewer 1 and 2, the paper will need to go to reviewers for re-evaluation if you chose to submit a revision.

Essential revisions:

1) A better link should be made between what Reviewer 1 identifies as the two parts of the manuscript (scRNAseq and the separate quantification of the septal neurons).

2) Particular attention to Reviewer 1 points 1-6, in particular the classes and types of cells being quantified based upon their label, distinguishing between neuron and oligodendrocyte, and better definition of the 'septum proper' and the 'septal eminence'.

Non-essential but would be an improvement

Cross-species analysis, ideally with human.

*Reviewer #1:*

In the first part of the manuscript the authors use single cell RNA sequencing to profile septal cells from embryogenesis to adult stages in order to capture the developmental trajectories of septal neurons and unravel their genetic codes. In the second part, the authors use transgenic mice to quantify septal cells derived from two septal progenitor zones. Finally, the authors examine the temporal development of septal cells generated from one of these zones and classify the neurons according to morphology.

It is an interesting manuscript with novel findings. In particular, single cell sequencing of the developing and adult septum is an exciting advance and one that is long overdue. However, one weakness is that, in its present form, the manuscript does not deliver the clarity in developmental progression with respect to the multiple populations of septal neurons that have already been identified and characterised functionally in the literature.

In the second part of the story, where cells derived from two progenitor zones of the septum are being quantified, there is no distinction made between neurons and glial cells hence does not provide a clear view of neuronal types generated from these progenitor zones.

This is an exciting paper but can be strengthened substantially if a link was made between the two parts of the study. For example, the authors could identify and show in the first part of the story potential lineage transcriptomes of progenitors and neurons derived from the two progenitor zones that they explored in the second part of the story. This will provide the reader with potential lineage-restricted transcriptomes.

1. There are a number of issues in the second part of the manuscript:

a. In figure 3H it is unclear what the authors are counting and in which mouse. Have sections been stained with a ZIC4 antibody? The mice used for the counts in G, H and I should be indicated in the figure.

b. In figure 3H and I, the authors are counting total ZIC4+ (unclear how these are identified) and total CELLS instead of neurons. And yet they go on to conclude on page 13 and 14 that using their intersectional approach they 'confirm that the vast majority of Nkx2.1-lineage NEURONS in the MS and LS are derived from the septal eminence rather than from the MGE/PoA, since these regions do not express Zic4'. As far as I can see, they did not count neurons but total cells and these include all the glia which undoubtedly make a large component of the cell counts. In addition, when they did count NeuN+ve cells in the Zic4-cre mice, they clearly show that only 50% of MS neurons are derived from septal Zic4 progenitors. So, in fact, 50% of MS neurons remain unaccounted for.

c. In the same figure, the authors refer to Figure 3F,I and state that 'the vast majority of labelled cells in the septum, with the exception of an abundant MS astrocyte population, were positive for GFP, confirming their origin in the septal eminence'. Where is this shown? There is no labelling with an astrocyte marker. In addition, none of the graphs in 3I show labelling higher than 50%.

d. The graphs in figure 3 are too small and difficult to read.

e. All the graphs in figure 3I have an n=2.

f. In figure 4E the authors quantify tdTomato cells labelled upon tamoxifen induction and use this to discuss the pattern of neuronal emergence in the septum. However, as far as I can see, they did not distinguish between neurons and cells of the oligodendrocyte lineage which would be labelled in this transgenic: Nkx2.1 is expressed in ventricular zone cells and Ascl1 is known to label oligodendrocyte precursors in the forebrain (see https://doi.org/10.1523/JNEUROSCI.0126-07.2007).

2. From the outset, the authors refer to the 'septum proper' and the 'septal eminence' and indicate that these represent 'two portions of the embryonic brain' (page 4). As far as I am aware, such a subdivision of the septum has not been defined. There are no anatomical landmarks to make such a division. These two 'portions' simply represent septal precursors that express different molecular identifiers which closely resemble the LGE or the MGE. The authors should either define these domains here or refer to them as anterior-dorsal and posterior-ventral domains, respectively. Or, if they have been defined in the literature, they should refer to that work.

3. On page 4 the authors state that the septum follows a general 'inside-out' pattern of neurogenesis. This is misleading. The cerebral cortex follows an inside-out neurogenesis and this refers to neurons close to the ventricular zone being generated first, and subsequent neurons having to migrate through pre-existing neurons to reach their destinations. In the case of the septum, and as described in Wei et al., 2012, neurons are generated in an outside-in manner.

4. On page 12 the authors state that 'certain types of neurons, such as those derived from Nkx2.1-expressing progenitors in the septal eminence, might follow different patterns' -referring to the 'inside-out' model and their suggestion that the appearance of these neurons resembles a fountain-like distribution. However, looking at their Figure S6C, it is clear that early-born neurons occupy more medial septal positions compared to later-born ones which are closer to the lumen. This distribution is not unlike what one would expect from the already proposed MS-to-LS birthdate or the outside-in model already proposed (http://dx.doi.org/10.1016/j.neuroscience.2012.07.016).

5. The authors should state how many cells originated from each developmental stage from the total of 72243 cells used across the six developmental stages.

6. In figure 2D the E12 and E14 images appear to have been swapped.

*Reviewer #2:*

This is an excellent description of septal development in mouse, and area that has not been studied with these cutting edge genetic lineage tracing and scRNA seq methods.

Is it known that all lhx6+ cells in the septal nucleus originate in the MGE? Please clarify.

Some cross-species analysis of human would greatly improve the paper.

[Editors' note: further revisions were suggested at acceptance, as described below.]

Congratulations, we are pleased to inform you that your article, "Transcriptional profiling of sequentially generated septal neuron fates.", has been accepted for publication in eLife.

*Reviewer #1:*

The revised version is much improved. However, the manuscript still fails to deliver the excitement of a single cell sequencing study of the septum across development and in the adult. The different parts of the study have not been adequately linked to create a coherent story. As is, the manuscript explores some single cell sequencing, some fate-mapping and some morphological analysis and makes no attempt to link these parts among them or, in some cases, to the literature. For example, since the authors are unable or unwilling to extract lineage-restricted transcriptomes from the two progenitor zones that they explored in the second part of the study, they could at least use existing single cell sequencing databases for the developing MGE, to extract MGE versus 'septal eminence' transcriptomes to provide an interesting insight into the septum, given that their dissections included the MGE. Or even extract the septal eminence versus septum proper transcriptomes. As they stand, the sequencing data represent lists and lists of genes without adequate impactful biological insight and indeed without distinguishing MGE from septal populations. As the authors say, 'our scRNA-Seq dataset can be used to identify diverse molecular cell types and infer their developmental trajectories within the developing septum'. This is what the authors should have done instead of launching into other aspects of septal development, none of which end up being adequately explored - referring to fate-mapping and morphological work.

There are several other issues:

The authors use ZIC labelling in some of their fate-mapping instead of using NEUN in all the panels. It is unclear why this has been done. We have no information as to the co-localization between NEUN and ZIC in the adult so it is unclear what we can conclude from this.

Staining for ASCL1 seems non-specific: ASCL1 is nuclear and from these tiny images, it seems to be cytoplasmic.

Some claims that are not really supported by the data:

'As far as we are aware, this is the first report of a mouse line that grants wide genetic access to GABAergic neurons in the lateral septum': this cannot be concluded from the low resolution image in Fig 1- fig supp 3.

'we found a stream of ZIC-positive cells migrating rostrally from their putative site of origin in the caudal septum': do they mean tdTomato cells migrating? Because this is what the image shows and not ZIC4 cells migrating.

Some of the references to Figures in the text are incorrect.

---

## [Author Response]

Reviewer #1:This is an exciting paper but can be strengthened substantially if a link was made between the two parts of the study. For example, the authors could identify and show in the first part of the story potential lineage transcriptomes of progenitors and neurons derived from the two progenitor zones that they explored in the second part of the story. This will provide the reader with potential lineage-restricted transcriptomes.

We thank Reviewer #1 for their considerations. Based on them, we felt that we had not managed to convey our reasoning well enough in the previous version of our manuscript. We have now changed the order in which the data is presented, to facilitate the understanding of our conceptual choices. As in the previous version of the manuscript, we begin by presenting our methodology and main scRNA-Seq findings in Figures 1 and 1-S1–1-S3. We then decided to split former Figure 3 into its embryonic (Figure 2) and mature (Figure 3, 3-S1) aspects in order to improve the flow of the paper and center the story around the *Nkx2.1* lineage. In the current version of our manuscript, we first describe the septal eminence as a unique proliferative region (Figure 2). We follow this with our fate-mapping experiments (Figures 3, 3-S1) to demonstrate the relevance of the *Nkx2.1* lineage within the mature septum. We then explore the potential transcriptomic changes taking place during embryonic development to guide the generation of early-born *versus* late-born (roughly MS *vs*. LS) neurons in both the septum and the septal eminence (Figures 4, 4-S1), and finish, as before, with our study of temporal cohorts of *Nkx2.1*-lineage neurons as a way to integrate our scRNA-Seq data (temporal changes in transcriptomic programs) with fate-mapping of cells born at different developmental stages. We very much look forward to making our dataset fully available to the public, so that readers can explore the lineages they are most interested in, hopefully expanding on our findings.

1. There are a number of issues in the second part of the manuscript:a. In figure 3H it is unclear what the authors are counting and in which mouse. Have sections been stained with a ZIC4 antibody?

We appreciate this comment and realize that it was insufficiently explained in the text. We have included a brief explanation in page 7, which we can expand on here: We used an antibody that recognizes several ZIC isoforms, including ZIC4 (see Borghesani *et al.*, Development 2002); this antibody has been used in other studies as a proxy for several ZIC isoforms (see Tiveron *et al.,* J Neuroscience 2017). Given the high overlap between the expression patterns of most of these genes in the septum (which can be verified using the Allen Brain Atlas; see also Inoue *et al.,* J Neuroscience 2007), we are confident that a pan-ZIC antibody can be used as a good readout for the number of ZIC4-positive cells (or their progeny) present in the tissue.

The mice used for the counts in G, H and I should be indicated in the figure.

We have changed the figure to include the corresponding mouse line at the top of both the representative image and analysis, which are now framed together for increased readability (i.e. Figure 3A+B; Figure 3C+D and Figure 3E+F). We have modified the panel abelled order and the figure legend accordingly to make this easier to follow.

b. In figure 3H and I, the authors are counting total ZIC4+ (unclear how these are identified) and total CELLS instead of neurons. And yet they go on to conclude on page 13 and 14 that using their intersectional approach they ‘confirm that the vast majority of Nkx2.1-lineage NEURONS in the MS and LS are derived from the septal eminence rather than from the MGE/PoA, since these regions do not express Zic4’. As far as I can see, they did not count neurons but total cells and these include all the glia which undoubtedly make a large component of the cell counts. In addition, when they did count NeuN+ve cells in the Zic4-cre mice, they clearly show that only 50% of MS neurons are derived from septal Zic4 progenitors. So, in fact, 50% of MS neurons remain unaccounted for.

We are grateful for Reviewer #1 for pointing this out. The counts in Figure 3D (formerly Figure 3H) do indeed contain both neurons and glia, but we have now included a new set of samples in Figure 3F, where we quantified NeuN positivity within each of the abelled populations, and can now be sure that the vast majority of LS neurons within the *Nkx2.1*-lineage are derived from the septal eminence. Separating former Figure 3 into current Figures 2 and 3 allowed us to clarify this point by bringing it forward explicitly in the title of Figure 3 and its corresponding section in the Results. Since we found Reviewer #1’s “cells vs. neurons” statement brought up a very interesting topic, we thought that it merited further discussion; we have now included considerations about glia, both within our dataset and as previously discussed in Wei *et al.,* 2012; this can be found in Figure 1-S2A-C and pages 5, 9 and 10. We also realized that our previous explanation of the intersectional model and justification for its use was clearly insufficient, so we have now extended and refined it both in the Results and in the Discussion sections (pages 9 and 15). This will hopefully help to clarify that while about 50 % of MS neurons are not originated from the developing septum (i.e. do not belong to the *Zic4*-lineage), the origin of part of those cells is in the PoA (this hypothesis is supported by the Shh-Cre fate-mapping data in Wei *et al.,* Neuroscience 2012), and/or in the MGE; this is in line with the presence of a small but consistent neuronal population in the MS that we describe in our study (Figures 3F, 3-S1C,E), as we discuss in page 15. There is, however, another portion of MS neurons, comprising at least half of the PV+ and VGluT2^+^ populations, whose developmental origin remains unknown, since they do not belong to either the *Nkx2.1*- or *Zic4*-lineages (Magno *et al.,* Cell Reports 2017). Since the focus of our manuscript is on the septal eminence, and our data do not provide any new information about these cells, we chose not to discuss them in the main text. We are grateful for the opportunity to clarify this here.

c. In the same figure, the authors refer to Figure 3F,I and state that ‘the vast majority of labelled cells in the septum, with the exception of an abundant MS astrocyte population, were positive for GFP, confirming their origin in the septal eminence’. Where is this shown? There is no labelling with an astrocyte marker. In addition, none of the graphs in 3I show labelling higher than 50%.

As outlined above, our new data include immunofluorescence staining for NeuN (Figures 3F, 3-S1B-E), as well as close-up images of the MS and the LS, where it is easier to appreciate the morphology of reporter-labeled cells than it is in Figure 3E (former Figure 3F). While it is true that we used no specific astrocyte markers, readers will be able to appreciate the clear astrocytic morphology of most cells within the subtractive population in the MS. We have also included a mention of this fact in the text in pages 9 and 15, while also referring to the Wei *et al.,* Neuroscience 2012 study, where they describe a similar observation. We realize the data as previously presented in former Figure 3I was not very informative (% of GFP+ cells allocated to each septal nucleus; this was the reason why the percentages shown were relatively low, as none of the septal nuclei harbored more than 50% of the intersectional population); we present a more complete picture in current Figure 3F, which now includes a better visualization of the same data, incorporating a similar analysis for the subtractive population, as well as the proportion of NeuN+ cells within each population and nucleus, which is further detailed in Figure 3-S1D,E.

d. The graphs in figure 3 are too small and difficult to read.

We have now separated former Figure 3 into current Figures 2 and 3. This allowed us to increase the size of the graphs within Figure 3, hopefully improving their readability.

e. All the graphs in figure 3I have an n=2.

We appreciate this comment, as we were aware that it was a weakness in the former version of the manuscript. We have obtained a new dataset including immunofluorescence staining for NeuN, where N=4 (Figures 3F, 3-S1B-E). While the general conclusions remain consistent with the data we had included in former Figure 3I, it is now more informative and comprehensive.

f. In figure 4E the authors quantify tdTomato cells labelled upon tamoxifen induction and use this to discuss the pattern of neuronal emergence in the septum. However, as far as I can see, they did not distinguish between neurons and cells of the oligodendrocyte lineage which would be labelled in this transgenic: Nkx2.1 is expressed in ventricular zone cells and Ascl1 is known to label oligodendrocyte precursors in the forebrain (see https://doi.org/10.1523/JNEUROSCI.0126-07.2007).

Reviewer #1 raises an interesting point. Based on the Parras *et al.,* J Neuroscience 2007 study, we would expect approximately 35 % of OPCs in the late embryonic (E17) septum to either express *Ascl1* or be derived from *Ascl1*-expressing cells. However, it is unclear how many of adult OPCs or mature oligodendrocytes in the septum belong in the *Ascl1*-lineage (while they did analyze 6 weeks-old samples, they do not report any numbers for the septum). Other *Ascl1* fate-mapping studies (such as Kim *et al.,* Mol Cell Neuroscience 2008) do not clarify this issue either. Nevertheless, our work in Figure 5 (former Figure 4) and 5-S1 focuses exclusively on *Nkx2.1*-lineage cells, and Kessaris *et al.,* (Nat Neurosci 2006) established that oligodendrocytes and their progenitors within this lineage are largely replaced by other lineages by P30, the age we examined in Figures 5 and 5-S1. This is fully consistent with our observations, since the vast majority of the cells we detected had a distinct neuronal morphology (contradicting our initial expectation that we would find glial cells as well). The reasons for this are beyond the scope of our study, but we can hypothesize that the our sparse abelling strategy might only be able to efficiently label Ascl1^high^ cells, as they approach M-phase (Imayoshi *et al.,* Science 2013), and that these might largely correspond to neurogenic progenitors in the septal eminence, which does not seem to be a major source of non-neuronal cells in the mature septum (see % of NeuN+ cells within the intersectional population, Figure 3F and 3-S1D – of course, it is also possible that a portion of the neurons labeled in Figure 5 could belong to the subtractive population, but in that case we would expect them to be within the small NeuN+ fraction). It is worth noting that fate-mapping using Ascl1-CreER shown by Kelley et al., 2018, found that M-phase progenitors comprised the major abelled population during embryonic induction up until E16.5 after which was a sharp decline which coincided with the switch to gliogenesis.

2. From the outset, the authors refer to the ‘septum proper’ and the ‘septal eminence’ and indicate that these represent ‘two portions of the embryonic brain’ (page 4). As far as I am aware, such a subdivision of the septum has not been defined. There are no anatomical landmarks to make such a division. These two ‘portions’ simply represent septal precursors that express different molecular identifiers which closely resemble the LGE or the MGE. The authors should either define these domains here or refer to them as anterior-dorsal and posterior-ventral domains, respectively. Or, if they have been defined in the literature, they should refer to that work.

Again, this is a very valid point. The previous version of our manuscript did not dedicate enough attention to this issue, and did not make a clear statement about why we chose the name “septal eminence” to refer to the caudal portion of the septum. We have now included a short review of previous literature referring to this structure and given a detailed explanation for its proposed renaming in the Discussion (pages 13 and 14); we also include a clarification in the Results section as well (pages 7 and 8). We believe that the term “septal eminence” is an adequate descriptor that will hopefully be adopted by the community.

3. On page 4 the authors state that the septum follows a general ‘inside-out’ pattern of neurogenesis. This is misleading. The cerebral cortex follows an inside-out neurogenesis and this refers to neurons close to the ventricular zone being generated first, and subsequent neurons having to migrate through pre-existing neurons to reach their destinations. In the case of the septum, and as described in Wei et al., 2012, neurons are generated in an outside-in manner.

This is true. We chose the term “inside-out” in a purely anatomical term, whereby early-born neurons are located towards the “inside” of the brain (i.e. on medial positions such as the medial septum), and later-born ones occupy progressively more lateral (or “outside”) positions. As Reviewer 1 points out, the “outside-in” usage in Wei *et al.,* Neuroscience 2012 is in opposition to the classic “inside-out” layering of the cortex. We see how our chosen terminology was confusing, and have changed it to “medial-to-lateral” (following the descriptive term first employed in Creps J Comp Neur 1974) in every instance, to still reflect the anatomical considerations we wanted to make while avoiding creating further confusion.

4. On page 12 the authors state that ‘certain types of neurons, such as those derived from Nkx2.1-expressing progenitors in the septal eminence, might follow different patterns’ -referring to the ‘inside-out’ model and their suggestion that the appearance of these neurons resembles a fountain-like distribution. However, looking at their Figure S6C, it is clear that early-born neurons occupy more medial septal positions compared to later-born ones which are closer to the lumen. This distribution is not unlike what one would expect from the already proposed MS-to-LS birthdate or the outside-in model already proposed (http://dx.doi.org/10.1016/j.neuroscience.2012.07.016).

We have addressed Reviewer #1’s concern in the text, acknowledging the general medial-to-lateral gradient before explaining the nuances specific to the *Nkx2.1* population (i.e. the “fountain-like pattern”). It is worth noting that in the Wei *et al.,* Neuroscience 2012 study, the authors note that “the birth sequence of septal neurons does not strictly correspond with the medial–lateral five-layer organization”, based on the patterns they observe for certain molecularly defined populations (CR+ in LSv are born later than CB+ neurons in LS, and ChAT+ neurons in MS2 are born later than nNOS+ neurons in MS3); our results reflect this discrepancy within the neurons of a specific developmental lineage (*Nkx2.1*).

5. The authors should state how many cells originated from each developmental stage from the total of 72243 cells used across the six developmental stages.

This was already indicated for the graphs displayed in Figure 1 and 1-S3, but we have now included the number of cells that passed the initial quality filters for every stage in the Methods section (page 21).

6. In figure 2D the E12 and E14 images appear to have been swapped.

This is true – we really appreciate the chance to correct that mistake.

Reviewer #2:This is an excellent description of septal development in mouse, and area that has not been studied with these cutting edge genetic lineage tracing and scRNA seq methods.Is it known that all lhx6+ cells in the septal nucleus originate in the MGE? Please clarify.

Reviewer #2 raises an important point. Within the ventral forebrain, *Lhx6* has been considered as a marker specific to the MGE ever since its expression patterns were first reported in that region (Grigoriou *et al.,* Development 1998). As we show in the Allen Brain Atlas in situ images in Figure 1G,H, there is a stream of cells expressing *Lhx6* that appear to be migrating in a caudal-to-rostral direction from the MGE into the septum, visible at E14 (Figure 1H) but not at E11 (Figure 1G). We assume that those cells, which coexpress Nkx2.1 but not Zic4 (this can be seen by comparing the SPRING plots in Figures 1H and 2A), are derived from the MGE, as the proliferative zone immediately adjoining the septum. Besides the MGE, *Lhx6* is expressed in the hypothalamus, where it only partially overlaps with *Nkx2.1* expression in the intrahypothalamic diagonal domain (Shimogori *et al.,* Nat Neurosci 2010). A recent study demonstrates that a subgroup of septal cells are originated from the diencephalon (Watanabe *et al.,* Scientific Reports 2018); as *Lhx6* is a crucial element in the specification of GABAergic hypothalamic neurons, in a region relatively close to the septum (Kim et al., Commun Biol 2021), it is reasonable to wonder if part of the *Lhx6* population that we captured in our scRNA-Seq dataset might be diencephalic in origin. However, we discard that possibility for two main reasons: 1) The thalamic eminence (TE)-derived cells are glutamatergic, and thus unlikely to coincide with an *Lhx6*-epxressing population; and 2) cells expressing *Calb2*, a marker of the TE-derived septal neurons, are largely separated from those expressing *Lhx6* in our SPRING plot.

Some cross-species analysis of human would greatly improve the paper.

We fully agree with Reviewer #2. However, the difficulty in obtaining human samples at comparable developmental stages makes it unfeasible to tackle such a task in a reasonable amount of time to add to the current study.

[Editors' note: further revisions were suggested at acceptance, as described below.]

Reviewer #1:The revised version is much improved. However, the manuscript still fails to deliver the excitement of a single cell sequencing study of the septum across development and in the adult. The different parts of the study have not been adequately linked to create a coherent story. As is, the manuscript explores some single cell sequencing, some fate-mapping and some morphological analysis and makes no attempt to link these parts among them or, in some cases, to the literature. For example, since the authors are unable or unwilling to extract lineage-restricted transcriptomes from the two progenitor zones that they explored in the second part of the study, they could at least use existing single cell sequencing databases for the developing MGE, to extract MGE versus 'septal eminence' transcriptomes to provide an interesting insight into the septum, given that their dissections included the MGE. Or even extract the septal eminence versus septum proper transcriptomes. As they stand, the sequencing data represent lists and lists of genes without adequate impactful biological insight and indeed without distinguishing MGE from septal populations. As the authors say, 'our scRNA-Seq dataset can be used to identify diverse molecular cell types and infer their developmental trajectories within the developing septum'. This is what the authors should have done instead of launching into other aspects of septal development, none of which end up being adequately explored - referring to fate-mapping and morphological work.

We appreciate the words of Reviewer #1, and are glad to read that they find the manuscript to be improved. We agree with this, and since most of the improvement was motivated by Reviewer #1’s comments to our initial submission, we are sorry to hear that they do not find our reworking of the manuscript satisfactory enough. We do, however, take on board their suggestion that we compare transcriptomes of defined groups of progenitors, and will strive to do so in the near future. Indeed, we hope that our dataset will be of use to any other researchers interested in this type of comparative approach, and look forward to its investigation by other groups.

There are several other issues:The authors use ZIC labelling in some of their fate-mapping instead of using NEUN in all the panels. It is unclear why this has been done. We have no information as to the co-localization between NEUN and ZIC in the adult so it is unclear what we can conclude from this.

We believe that the main concern expressed by Reviewer #1 in this point is that immunostaining for ZIC (Figure 3D,E) might underestimate the number of cells from this lineage present in the adult septum, since some of them might have lost the expression of *Zic* genes along development. This, in turn, might skew the quantification of Nkx2.1-lineage cells that we present in Figure 3E. We agree that the ideal scenario would be to present a quantification of how many tdTomato+ cells are also positive for NeuN; unfortunately we had purely practical reasons for not performing this experiment as we did not have readily available samples within this mouse line. However, we found that the numbers we presented were still informative. As shown in Figures 3A and B, the vast majority of neurons (i.e. NeuN+ cells) have a developmental history of *Zic4* expression, especially in the lateral septum; this means that the denominators in Figure 3D can be approximated to the overall neuronal populations. Admittedly, this is not clear-cut in the medial septum, where -as pointed out by Reviewer #1 in their initial comments- only about half of the neurons belong to the Zic lineage. We are intrigued by this and would like to investigate the nature and developmental origin of these neurons further.

Staining for ASCL1 seems non-specific: ASCL1 is nuclear and from these tiny images, it seems to be cytoplasmic.

We understand reviewer #1’s concern, which touches upon our own when submitting images at low resolution as mandated by journal specifications. It is true that the ASCL1 antibody that we used can result in non-specific binding, especially in blood vessels (since it is raised in mouse) and in the ECM-rich subventricular zones of the subpallium (which is one of the reasons why we limited our study to the ventricular zone). However, we do have previous experience using this particular reagent, and we are confident that we obtained a good set of immunostainings – hopefully this will be easier to see in the example image in Author response image 1 (single channel image, with ASCL1 in gray, as shown in Figure 2F, Nkx2.1+ area), where blood vessels (red asterisks) appear as bright features, and individual nuclei are clearly distinguishable.

**Author response image 1. sa2fig1:** 

Some claims that are not really supported by the data:'As far as we are aware, this is the first report of a mouse line that grants wide genetic access to GABAergic neurons in the lateral septum': this cannot be concluded from the low resolution image in Fig 1- fig supp 3.

We include a bigger version of panel 1S3E in Author response image 2, which will hopefully make it easier to appreciate how many of the neurons in the lateral septum (but not in the medial septum) are labeled by tdTomato, supporting the statement we make in our manuscript (“Nearly all labeled cells were neurons confined to the lateral septum (Figure 1 – figure supplement 3E), and thus could be assumed to be largely GABAergic (Zhao *et al*., 2013)”).

'we found a stream of ZIC-positive cells migrating rostrally from their putative site of origin in the caudal septum': do they mean tdTomato cells migrating? Because this is what the image shows and not ZIC4 cells migrating.

This is a very good point, given that in Figure 2C we only show low-magnification images, making it hard to distinguish individual cells stained for both tdTomato and ZIC. We include in Author response image 3 a panel from the drafting stage of our manuscript, cropped from the area shown in Figure 2D, where many of the tdTomato+ (green) cells with migratory morphology are co-stained with the pan-ZIC antibody (magenta).

**Author response image 3. sa2fig3:** 

Some of the references to Figures in the text are incorrect.

This was true, and mainly due to our re-labeling of the panels in Figure 3 in hopes of improving the flow of data. We have hopefully located and corrected all the mistakes.